

# Decadal-scale decay of landslide-derived fluvial suspended sediment after Typhoon Morakot

Gregory A. Ruetenik[1], Ken L. Ferrier[2], and Odin Marc[3]

[1]Institute of Geophysics, Czech Academy of Sciences, Prague, Czech Republic
[2]Department of Geoscience, University of Wisconsin-Madison, Madison, WI, USA
[3]Géosciences Environnement Toulouse (GET), UMR 5563, CNRS/IRD/CNES/UPS, Observatoire Midi-Pyrénées, Toulouse, France

**Correspondence:** Gregory Ruetenik (ruetenik@ig.cas.cz)

**Abstract.**

Landslides influence fluvial suspended sediment transport by changing sediment supply and grain size, which alter suspended sediment concentrations and fluxes for a period of time after landsliding. To investigate the duration and scale of altered suspended sediment transport due to landsliding, we analyzed suspended sediment concentration and water discharge measurements at 87 gauging stations across Taiwan over an 11-year period after Typhoon Morakot, which generated nearly 20,000 landslides in 2009. At each gauging station, we computed annual rating curves to quantify changes over time in the sensitivity of suspended sediment concentrations to water discharge. Among the 40 stations in basins that were impacted by landsliding, the discharge-normalized rating curve coefficient $\tilde{a}$ was higher than that before Morakot by a factor of $5.1 \pm 1.1$ (mean $\pm$ standard error) the first year after Morakot (2010). The rating curve exponent $b$ did not decrease at most stations until a year later (2011), when the average $b$ value was lower than that before Morakot by $0.25 \pm 0.05$. Across the compilation of gauging stations, post-Morakot changes in $\tilde{a}$ were positively correlated with landslide intensity for seven years after Morakot, while post-Morakot changes in $b$ were negatively correlated with landslide intensity from 2011 to 2014, reflecting a tendency for larger changes in $\tilde{a}$ and $b$ to occur in basins with more intense landsliding. At 26 of these 40 stations, elevated values of $\tilde{a}$ declined after the initial post-Morakot peak, consistent with a gradual return to pre-Morakot suspended sediment transport conditions. Exponential regressions to these $\tilde{a}$ values reveal a median characteristic decay time of 8.8 years (interquartile range: 5.7-14.8 years). Values of $\tilde{a}$ tended to decline faster in basins with more intense landsliding, with a mean characteristic decay time of 6 years in the basins hardest hit by landsliding. Shortly after Morakot, changes in $\tilde{a}$ and $b$ tended to be larger in basins with more intense landsliding, but this sensitivity to landsliding decayed away within 4-7 years. At stations that were not impacted or only minimally impacted by landsliding, neither $\tilde{a}$ nor $b$ exhibited systematic responses to Morakot. To quantify the effect of landsliding on sediment discharge, we compared the measured sediment discharges after Morakot to the hypothetical sediment discharges that would have occurred if Morakot had induced no landslides, calculated by applying each station's pre-Morakot rating curve to its post-Morakot water discharge history. This analysis suggests that Morakot-induced landsliding increased sediment discharge by as much as >10-fold in some basins in the 1-2 years after Morakot. Together, these results indicate that the influence of Morakot-induced landsliding on rating curves was large shortly after Morakot but diminished in less than a decade in most of the study rivers, and will be imperceptible in another few decades in all of the study rivers. To the





extent that these results are applicable to other landscapes, this suggests that periods of elevated sediment transport efficiency after landsliding should persist for years to decades, even if the landslide deposits persist for centuries to millennia.

## 1 Introduction

Widespread landsliding events, such as those caused by heavy rainfall (e.g., Milliman and Kao, 2005; Marc et al., 2018; Yamada
et al., 2012), can deliver large amounts of sediment to rivers and temporarily elevate suspended sediment concentrations and sediment discharge (e.g. Milliman and Syvitski 1992; Hovius et al. 2000; Hovius et al. 2011). In fluvial monitoring records, this is often characterized by a brief spike in suspended sediment concentrations followed by a protracted tail of elevated suspended sediment concentrations (e.g., Hicks et al. 2008). In a given river, the duration of the tail depends on the river's ability to transport the landslide-mobilized material, and hence on the river's transport capacity and the size of the landslide
deposit that's accessible to the river. Recent studies have inferred a wide range of recovery times for suspended sediment flux responses to landsliding, from years (e.g., Hicks et al. 2008; West et al. 2014; Croissant et al. 2017) to thousands of years (e.g., Yanites et al. 2010).

Questions about the persistent influence of mass wasting on sediment concentrations are particularly relevant in Taiwan, which experiences frequent landsliding due to large rainfall events and large earthquakes. Taiwan is susceptible to frequent
landsliding because it is undergoing rapid uplift in response to arc-continent collision between Eurasia and the Philippine Sea Plate (Suppe 1984) and is one of the most rapidly eroding places on Earth, with denudation rates exceeding 10 mm/yr in some areas (Dadson et al. 2003; Fox et al. 2014). In many mountainous regions in Taiwan, rock uplift rates outpace erosion rates by diffusive soil transport, and landslides are the dominant source of hillslope erosion (Hovius et al. 2000). For example, the 1999 magnitude 7.3 Chi-Chi earthquake caused over 20,000 landslides covering over 150 km$^2$ (Dadson et al. 2004). Similarly,
Typhoon Toraji in 2001 induced >30,000 landslides, which resulted in >175 Mt of suspended sediment discharge from the Choshui River (Dadson et al. 2005), equivalent to three times its annual average sediment load over the period from 1986 to 1999.

In this study, we focus on the responses to landslides induced by Typhoon Morakot (August 5-9, 2009). This was the wettest typhoon on record in Taiwan, generating > 3 m of rainfall in the south-central portion of Taiwan (Figure 1a) and
close to 20,000 landslides with a net area > 250 km$^2$ (Lin et al. 2011). This disturbance was such that it may have altered the regional microseismicity (Steer et al., 2020), and it resulted in amplified fluvial sediment fluxes in many basins across Taiwan (Kao et al., 2010). Huang and Montgomery (2013) documented fluvial responses to Morakot by analyzing measurements of suspended sediment concentration ($C$) and water discharge ($Q$) at 19 gauging stations in southern Taiwan monitored by Taiwan's Water Resources Agency. With these data, they calculated two rating curves of the form $C = aQ^b$ (where $a$ and $b$
are constants) at each station: one for the monitoring period before Morakot, which spanned several decades, and another for the two years of post-Morakot measurements that had been made by that time. This revealed that the centered rating curve coefficient $a$ increased and the rating curve exponent $b$ decreased after Morakot at 15 of the 19 study stations, indicating more efficient suspended sediment transport at a given discharge and a smaller sensitivity of $C$ to $Q$.





Huang and Montgomery (2013) interpreted these changes in the rating curves as reflecting a shift from coarser to finer
sediment, which would generate a decrease in $b$ at the same time as an increase in $a$. This was supported by bed grain size
measurements before and after Morakot by the Water Resources Agency in the Beinan River, which revealed a reduction in
median grain size at this site after Morakot. Huang and Montgomery (2013) further observed that the magnitude of the grain
size fining in the Beinan River would be enough to shift the sediment transport regime from threshold bed (gravel) to live
bed (sand), which would increase sediment discharge at low flow. As Huang and Montgomery (2013) noted, although the
perturbations to the rating curves were large in many rivers, it was not possible to evaluate how long these perturbations would
last, since only two years of post-Morakot measurements were available at the time. If these perturbations persisted for long
times, then this would imply that large landslide events should generate enduring changes in the sediment transport regime,
sediment fluxes, and erosional unloading. If, on the other hand, these perturbations diminished quickly, then large landslide
events should only generate elevated sediment fluxes for geologically short times and have little influence on long-term mass
fluxes.

Recent advances offer an opportunity to reassess the duration of fluvial responses to Morakot and to quantify the sensitivity of
these perturbations to landsliding. Marc et al. (2018) generated a new inventory of Morakot-induced landslides across Taiwan,
which provides a comprehensive assessment of landslide volumes (Figure 1b). In addition, suspended sediment concentrations
and discharge have continued to be monitored at many rivers across Taiwan, with more than a decade of measurements now
accumulated since Morakot (Water Resources Agency, 2020).

Here we build on Huang and Montgomery (2013) to revisit two questions: 1) How much did Typhoon Morakot affect
fluvial suspended sediment fluxes at rivers across Taiwan? 2) How long did these effects last? To address these questions,
we analyzed suspended sediment concentration and water discharge measurements at 87 fluvial gauging stations in basins
across Taiwan (Water Resources Agency, 2020), which were affected by Morakot to varying degrees. Some of these basins
experienced intense landsliding during Morakot and showed major changes in suspended sediment transport after Morakot.
Others experienced no landsliding and showed no measurable change in suspended sediment transport after Morakot, offering
a baseline point of comparison for basins that experienced widespread landsliding. Collectively, these gauging station records
provide a means to quantify erosional responses to Morakot across Taiwan and a means to test the sensitivity of these erosional
responses to the intensity of landsliding.

This manuscript is structured around an analysis of the effects of Morakot-induced landsliding on fluvial sediment fluxes.
Section 2 summarizes the gauging station measurements and the methods we used to compute suspended sediment loads
and rating curves. Section 3 presents annual estimates of the rating curve parameters, suspended sediment loads, and basin-
averaged erosion rates, which reveal spatial and temporal variations in sediment fluxes across Taiwan in the decade after
Morakot. Section 4 discusses the duration of the perturbation to the rating curve parameters and their sensitivity to the intensity
of landsliding. Together, these results constrain the spatial and temporal extent of suspended sediment responses to Morakot-
induced landsliding, and they suggest that elevated suspended sediment concentrations in these basins will dissipate within a
few decades.





## 2 Methods

To quantify the duration of fluvial sediment fluxes affected by Typhoon Morakot, we analyzed monitoring records at 87 river
gauging stations (Figure 1a), all of which were operated by Taiwan's Water Resources Agency (WRA; Water Resources
Agency, 2020). Supplementary Table S1 lists each gauging station's WRA ID, river name, location, and monitoring duration.
The monitoring records at these stations are of variable length, with some dating back to 1948 and others starting as late
as 2017. The monitoring records consist of measurements of average water discharge at daily intervals and depth-averaged
suspended sediment concentrations at less frequent and irregular intervals (Kao et al., 2005). The median sampling interval for
suspended sediment concentrations across all gauging stations is 12 days during typhoon season (June-October) and 15 days
during the rest of the year.

As we describe below, we use the gauging station measurements to create rating curves relating suspended sediment con-
centration to water discharge, and we use these rating curves to compute suspended sediment discharge at each gauging station
during each year of each station's operation (Supplementary Tables S1-S5). In Sections 3-4 we present figures from a subset
of 24 stations (Figure 1a) that collectively span the range of responses across the full set of 87 stations, and which therefore
illustrate the sensitivity of fluvial suspended sediment fluxes to Morakot-induced landsliding. Twelve of the 24 focus stations
(labeled S1-S12 in Figure 1a) are in the southern half of the island, where precipitation during Morakot was highest and where
landsliding was most intense. In these basins, the volume of landslide-mobilized material per unit drainage area ranged from
440 to $2.7 \cdot 10^5$ m$^3$ km$^{-2}$. The other twelve stations (N1-N12) are farther north in Taiwan, where precipitation during Morakot
was less intense and landsliding was less common. Eight of these twelve basins had no Morakot-induced landsliding, and in
the remaining four basins, the volume of landslide-mobilized material per unit drainage area ranged from 4 to 34 m$^3$ km$^{-2}$.
These provide a baseline against which the responses at stations S1-S12 can be compared.

The fluvial monitoring records are of variable completeness. Temporal gaps are present in every gauging station's records,
with gaps as short as several days to as long as 50 years. On average across all the study gauging stations, 92% of the days
in the monitoring period have discharge measurements. For stations whose early records include large temporal gaps decades
before Morakot, we only compute rating curves in time periods with a minimum of five suspended sediment concentration
measurements. Among the 24 focus stations, the average number of suspended sediment concentration measurements per year
ranges from 15.0 to 29.7.



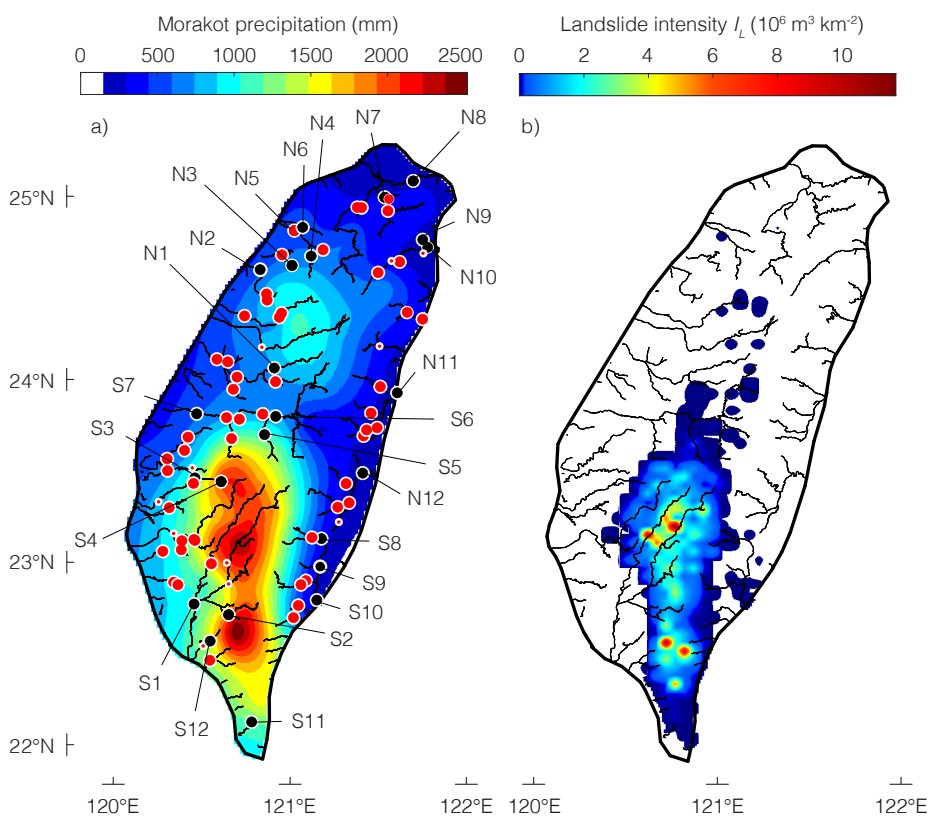

**Figure 1.** a) Locations of fluvial gauging stations analyzed in this study. The stations of focus in Figures 4, 5, 7, and 9 (black dots) are labeled S1-S12 in southern Taiwan and N1-N12 in northern Taiwan. Large red dots indicate stations used to calculate trends in rating curve parameters, and small red dots indicate stations used to compute sediment discharge (Table S1). Contoured colors represent rainfall totals during Typhoon Morakot derived from a kriging-based interpolation of rainfall gauging stations (Water Resources Agency, 2020). b) Spatial density of landsliding (volume of mobilized material per unit area) smoothed from the compilation in Marc et al. (2018).

## 2.1 Estimating rating curve parameters and suspended sediment loads

A river's suspended sediment load $Q_s$ [M T$^{-1}$] can be calculated as the product of volumetric water discharge $Q$ [L$^3$ T$^{-1}$], water density $\rho_w$ [M L$^{-3}$], and suspended sediment concentration $C$ [M M$^{-1}$] (Equation 1).

$$Q_s = CQ\rho_w \tag{1}$$

At the study gauging stations, $Q$ was measured every day but $C$ was measured less frequently, so direct measurements of $Q$ and $C$ cannot be used to calculate $Q_s$ every day. In such cases, a common approach is to estimate daily values of $C$ by applying

a power-law rating curve relating $C$ to $Q$ (Equation 2; e.g., Ferguson, 1986; Syvitski et al., 2000; Gao, 2008).

$$C = aQ^b \tag{2}$$





Here $b$ is a dimensionless constant that describes the sensitivity of $C$ to $Q$, and $a$ is a constant with dimensions of $\mathrm{T}^b\,\mathrm{L}^{-3b}$ that describes the concentration of suspended sediment at a given $Q$. At the study gauging stations, the reported measurements of $C$ are in units of parts per million (ppm), those of $Q$ are in units of $\mathrm{m}^3\,\mathrm{s}^{-1}$ (Water Resources Agency, 2020), and we assume
$\rho_w = 1000\ \mathrm{kg\ m}^{-3}$, so the units of $a$ are $\mathrm{ppm\ s}^b\,\mathrm{m}^{-3b}$.

Our goal is to quantify the influence of Morakot on suspended sediment transport by tracking the evolution of $Q_s$ and the rating curve parameters over time. To do this, we applied a commonly used methodology based on a modified version of Equation 2 to calculate annual estimates of the suspended sediment load and the rating curve parameters. This involved applying two steps before calculating values for the rating curve parameters.

The first step is centering the logged discharge data, which we did following Cohn et al. (1992). This reduces the influence of estimates of $b$ on estimates of $a$, which can confound efforts to compare rating curve parameters at different stations or different times (e.g., Syvitski et al., 2000; Warrick, 2015). This involved calculating the center of the $\log(Q)$ data with Equation 3— which we denote as $\mathrm{center}(\log(Q))$—and subtracting it from the $\log(Q)$ data over the time period of interest.

$$\mathrm{center}(\log(Q)) = \mathrm{mean}(\log(Q)) + \frac{\sum_{k=1}^{N}(\log(Q) - \mathrm{mean}(\log(Q))^3}{2\sum_{k=1}^{N}(\log(Q) - \mathrm{mean}(\log(Q))^2} \tag{3}$$

Here, $N$ is the number of discharge measurements and $\mathrm{mean}(\log(Q))$ is the mean of the log-transformed discharge data.

After centering the discharge, we estimated the rating curve parameters by applying the adjusted maximum likelihood estimation (AMLE) method of Cohn et al. (1989) to measurements of $\log(C)$ and the centered discharge $\log(Q) - \mathrm{center}(\log(Q))$. Graphically, this is analogous to a linear regression through log-transformed $C$ and $Q$ data normalized by $\mathrm{center}(\log(Q))$. The resulting parameter values are for the regression line's slope ($b$) and intercept evaluated at $\mathrm{center}(\log(Q))$. The intercept shares the same units as $C$ (here, ppm) and is denoted $\tilde{a}$ to distinguish it from $a$ in Equation 2. This is beneficial because it reduces the effects of the artifactual dependency of $a$ on $b$ that would have resulted from regressing $\log(C)$ against the uncentered $\log(Q)$ data, and thus simplifies comparison of $\tilde{a}$ values estimated at different times or stations. This method is identical to the AMLE method implemented in the Load Estimator computer program (LOADEST; Runkel et al. 2004), which is based on Cohn et al. (1989). To facilitate comparison of the rating curve parameter values among different stations and different times, we report
estimates of $\tilde{a}$ throughout this study.

We applied this approach to each year's $C$ and $Q$ data to compute annual estimates of the rating curve parameters, which permitted quantification of changes in the rating curves over time. To ensure that each year's estimate of $\tilde{a}$ at a given station can be straightforwardly compared to the other annual estimates of $\tilde{a}$ at the same station, we subtracted the same $\mathrm{center}(\log(Q))$ value from each year's $\log(Q)$ measurements at that station. To obtain a common value of $\mathrm{center}(\log(Q))$ to apply to each
year at a given station, we applied Equation 3 to all the discharge measurements over the gauging station's entire monitoring period. The resulting values of $\mathrm{center}(\log(Q))$ for each station are tabulated in Supplementary Table S2. The methodology is summarized in the example shown in Figure 2.

The second step we applied in these calculations is a correction for log transformation bias in estimates of $Q_s$. To make this correction, we followed the minimum-variance unbiased estimator method (MVUE) of Cohn et al. (1989). We used this
method and the daily measurements of $Q$ to calculate daily estimates of $Q_s$ corrected for log transformation bias. We used the



method of Gilroy et al. (1990) to estimate the uncertainty in the sediment load, following the implementation in LOADEST (Runkel et al., 2004). We calculated annual suspended sediment loads over each water year (November 1 - October 31) by summing the daily estimates of $Q_s$ over the year. For days without $Q$ measurements, we assigned the year's average daily $Q_s$ value.

After computing $Q_s$ at each gauging station, we computed basin-averaged erosion rates $E$ [L T$^{-1}$] by dividing $Q_s$ by the drainage area $A$ upstream of the gauging station and an assumed bedrock density $\rho_r$ of 2700 km m$^{-3}$.

$$E = \frac{Q_s}{\rho_r A} \tag{4}$$

These represent the spatially averaged erosion rate associated with the suspended sediment load. Because these do not include fluvial mass fluxes associated with bedload or dissolved loads, these values of $E$ represent a lower bound on total basin-averaged erosion rates.

Some basins contained multiple gauging stations along the same river. This yielded estimates of $Q_s$ in the catchment draining into the downstream gauging station as well as in the smaller catchment draining into the upstream gauging station, which is contained within the larger catchment. In these situations with nested catchments, we used Equation 5 to calculate $E$ for the portion of the large basin that isn't contained within the smaller tributary basin (e.g., Hu et al., 2021). For example, if we denote the area of the catchment draining into the downstream gauging station $A_1$, the area of the tributary catchment draining into the upstream gauging station $A_2$, and the suspended sediment loads from these stations $Q_{s1}$ and $Q_{s2}$, respectively, then the average erosion rate $E_3$ over the portion of $A_1$ that's not part of $A_2$ is:

$$E_3 = \frac{Q_{s1} - Q_{s2}}{\rho_r (A_1 - A_2)}. \tag{5}$$

To quantify year-to-year variations in sediment fluxes over time, we applied Equations 1-5 to each gauging station's measurements of $Q$ and $C$ during each water year. This yielded annual estimates of $\tilde{a}$, $b$, $Q_s$, and $E$ at each gauging station. We also applied this method to the entire period of $C$ and $Q$ measurements before Morakot at each gauging station to calculate the average pre-Morakot rating curve parameters, which we denote $\tilde{a}_{pre}$ and $b_{pre}$ (Supplementary Table S2). These serve as a baseline to compare post-Morakot values of $\tilde{a}$ and $b$ against. Lastly, we applied Equations 1-5 to two portions of the 2009 water year, one before Morakot (November 1, 2008 to August 4, 2009) and one after Morakot began (August 5 to October 31, 2009). This isolated the response to Morakot in the first few months after the typhoon from the portion of the 2009 water year that preceded the typhoon.





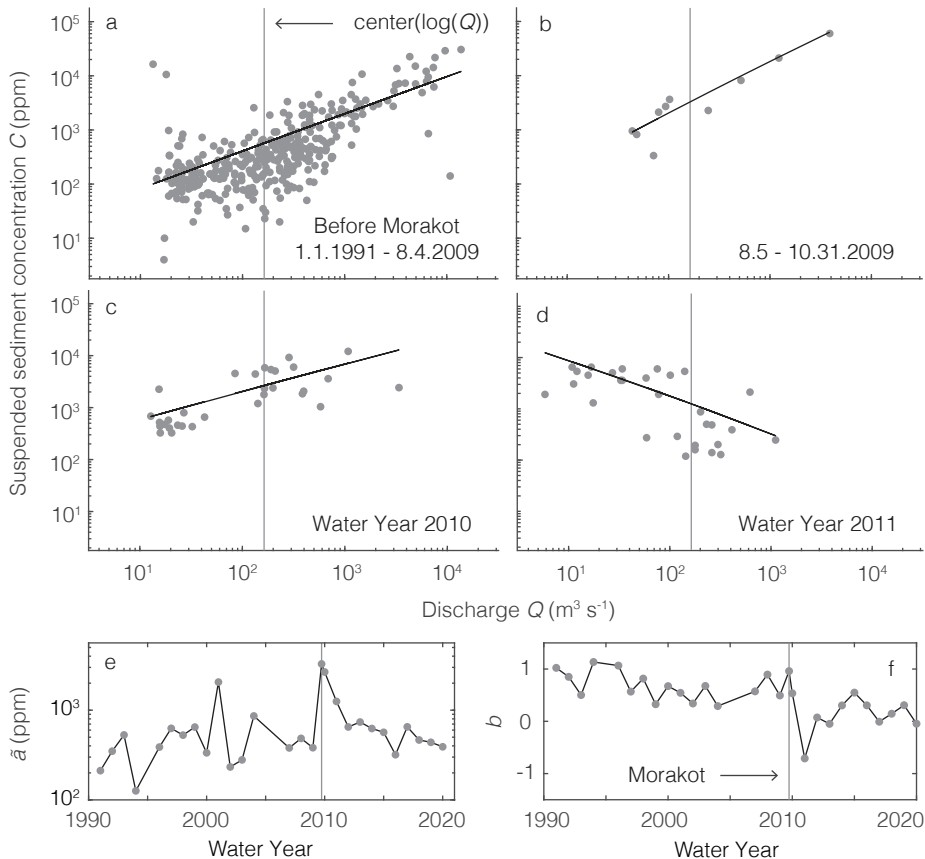

**Figure 2.** Example of measurements of suspended sediment concentration and water discharge before Morakot (panel a) and after Morakot (b-d) at station S1 in the Gaoping River (Figure 1; Water Resources Agency, 2020). Vertical lines show the center of the log-transformed discharge data over the entire monitoring period, which is the discharge at which the rating curve parameter $\tilde{a}$ is determined (Equation 3). **e, f**. Estimates of rating curve parameters $\tilde{a}$ and $b$ at this station from 1991 to 2020. These show the variability in annual estimates of $\tilde{a}$ and $b$ and the influence of Morakot on $\tilde{a}$ and $b$ at this station.

## 2.2 Estimating impacts of Morakot on sediment discharge

We aim to quantify the effects of Morakot on suspended sediment loads over the decade after Morakot. For rivers whose rating curves do not change during a given storm, it may be sufficient to apply the same rating curve to discharges before and after

the storm (e.g., Gao 2008). For other rivers, however, typhoons can alter a river's rating curve, such that the sediment load associated with a given water discharge differs before and after the typhoon (e.g., Kao et al. 2005; Hovius et al. 2000; Figure 3). In such cases, applying a river's pre-typhoon rating curve to post-typhoon discharge measurements would yield errors in estimates of post-typhoon suspended sediment concentrations and sediment discharge.

Estimates of a river's annual sediment load are sensitive not just to typhoon-induced changes in the rating curve parameters,

but also to the magnitude-frequency distribution of discharge that the river experiences after a typhoon (e.g., Kirchner et al.,



2011). For example, if precipitation happens to be lower the year after a typhoon than the year before it, then sediment loads may be lower the year after the typhoon than the year before it, even in rivers in which the rating curve coefficient is higher after a typhoon. Accounting for these temporal variations in discharge is particularly important in Taiwan, where precipitation rates are highly variable in time. For instance, cumulative rainfall in a given month in a given basin can vary by an order of

magnitude or more from year to year (Kao et al. 2005; Yu et al. 2006).

This implies that the effects of Typhoon Morakot cannot be determined by directly comparing a river's sediment loads before and after Morakot. Instead, to quantify the effect of Morakot on sediment loads, we compared two estimates of the annual sediment load. The first is determined with the conventional application of post-Morakot rating curves to the post-Morakot discharge history, as described in Section 2.1. We calculated a rating curve for each post-Morakot water year based

on the year's roughly biweekly $C$ measurements and concurrent $Q$ measurements (Figure 3a). Then, we applied each year's rating curve to that year's time series of daily $Q$ measurements to estimate daily $Q_s$ values (Figure 3b). Summing the daily $Q_s$ values over each year yielded our best estimates of the annual $Q_s$ (Figure 3c).

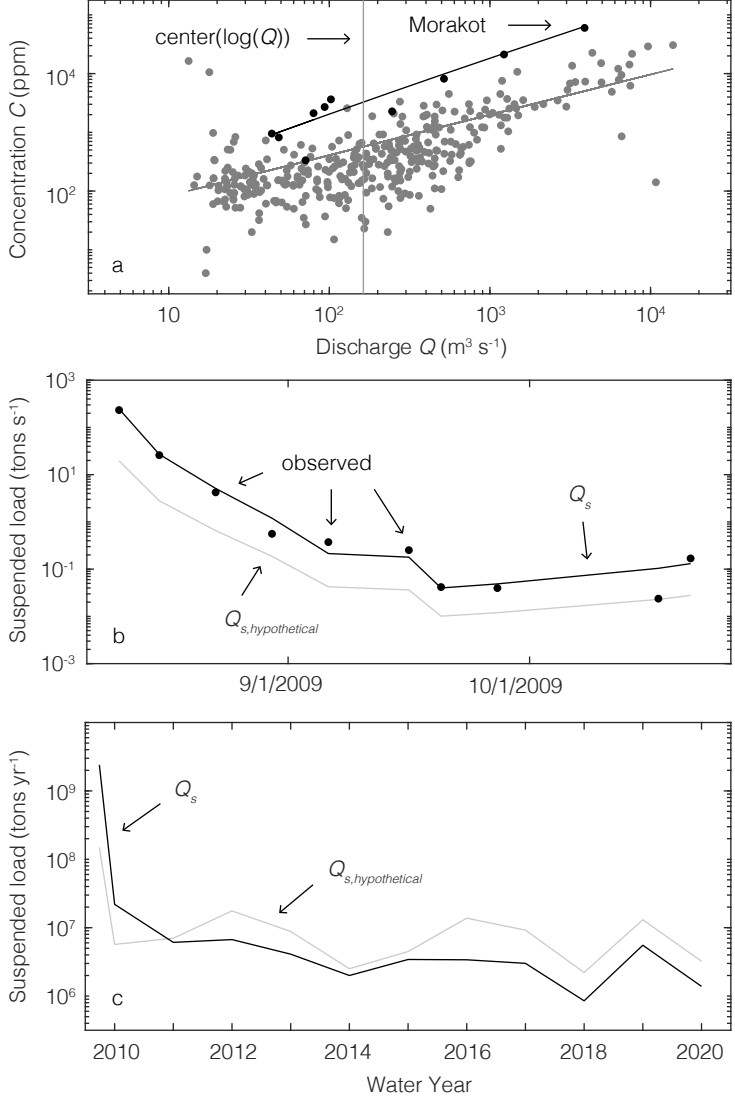

**Figure 3.** Example of approach for estimating effects of Typhoon Morakot on suspended sediment discharge $Q_s$. **a.** Concentration $C$ and discharge $Q$ measurements at station S1 in the Gaoping River (Figure 1) before Morakot (gray dots; Jan 1, 1990 – August 4, 2009) and during the portion of Water Year 2009 after Morakot began (black dots; August 5 – October 31, 2009). Regression lines are inferred rating curves for these time periods. Vertical line is at center($\log(Q)$) (Equation 3). **b.** Black dots show observed $Q_s$ on days with measurements of both $C$ and $Q$ during the post-Morakot portion of the 2009 water year. Black line shows $Q_s$ estimated by applying the post-Morakot rating curve (black line in panel a) to discharge measurements. Gray line shows the hypothetical $Q_s$ that would be obtained if the pre-Morakot rating curve (gray line in panel a) were applied to the same discharge measurements. Applying the pre-Morakot rating curve to the post-Morakot discharges would systematically underestimate sediment discharge. **c.** Sediment loads at S1 estimated by applying the pre-Morakot rating curve (gray line) and the annual post-Morakot rating curves (black line) to $Q$ measurements. At this site, $Q_s$ exceeds $Q_{s,hypothetical}$ for less than two years after Morakot.



The second estimate is a hypothetical suspended sediment load, $Q_{s,hypothetical}$, which is meant to answer the question: What would the post-2009 sediment loads have been if Morakot hadn't occurred but the rivers had experienced the same discharge

history that did occur after Morakot? We calculated $Q_{s,hypothetical}$ by applying the same methodology applied in Section 2.1 to the post-Morakot discharge history, except that each estimate used the pre-Morakot values $\tilde{a}_{pre}$ and $b_{pre}$, rather than computing new values of $\tilde{a}$ and $b$ each year.

We summed the daily estimates of $Q_{s,hypothetical}$ over each water year to obtain estimates of the annual $Q_{s,hypothetical}$ (Figure 3c). We take the ratio of the annual $Q_s$ and $Q_{s,hypothetical}$ to be a measure of Morakot's effect on annual suspended

sediment discharge.

## 2.3 Basin-averaged landslide intensity

To quantify the intensity of landsliding in drainage basins upstream of the gauging stations, we used landslide volumes in the inventory of Marc et al. (2018). In this study, Marc et al. (2018) mapped the areas of Morakot-induced landslide scars $A_L$ in Formosat 2 aerial imagery at 8-m multispectral (2-m panchromatic) resolution, and estimated landslide volume $V_L$ with

Equation 6 (Larsen et al., 2010).

$$V_L = cA_L^p \tag{6}$$

In the landslide catalogue in Marc et al. (2018), the calculation of $V_L$ included corrections for amalgamated landslide polygons (Marc et al. 2018), in which it is assumed that each landslide has an elliptical shape and a mean width calculated with the formula proposed and validated by Marc et al. (2018). These calculations also involved estimating scar area using a mean

length to width ratio derived from a global database of 277 landslide scars with volumes ranging from 1000 m³ to 1 km³ (Domej et al. 2017). In this catalogue, landslide volumes were calculated with Equation 6 with parameters for shallow landslide scars of $p = 1.262 \pm 0.009$ and $\log10(c) = $ -0.649 $\pm$ 0.021 and for bedrock landslide scars of $p = 1.41 \pm 0.02$ and $\log10(c) = $ -0.63 $\pm$ 0.06 (Larsen et al. 2010). We use the landslide volume estimates from Marc et al. (2018) directly.

We calculated the basin-averaged landslide intensity $I_L$ [L³ L⁻²] as the total upstream landslide volume (summed over all

landslides) divided by the drainage area $A$.

$$I_L = \frac{\sum V_L}{A} \tag{7}$$

## 3 Results

### 3.1 Rating curve parameters $\tilde{a}$ and $b$

At most of the southern stations, values of $\tilde{a}$ increased rapidly within one year after Morakot and then declined over the

following decade. At six of the eight southern stations with data in the post-Morakot portion of 2009 (S1-S2, S5-S7, S12), values of $\tilde{a}$ in the post-Morakot portion of 2009 were higher than $\tilde{a}_{pre}$ (the pre-Morakot average value of $\tilde{a}$) by an average factor of 3.4 (range 1.4-8.8). At the other two stations (S3-S4 in the Bazhang River basin), values of $\tilde{a}$ in the post-Morakot



portion of 2009 declined slightly to 0.7-0.8 times that of $\tilde{a}_{pre}$. At the remaining four southern stations (S8-10 in the Beinan River and S11 in the Sizhong River), no data was collected from June 2009 until January 2010, which prevents calculation of $\tilde{a}$

and $b$ values during this time. In water year 2010, the first time $\tilde{a}$ can be estimated at these four stations, values of $\tilde{a}$ are higher than $\tilde{a}_{pre}$ by an average factor of 14 (range 8.1-22).

By contrast, values of $\tilde{a}$ at the northern stations appear to be largely unaffected by Morakot (Figure 4). The average value of $\tilde{a}/\tilde{a}_{pre}$ across the twelve northern stations in the post-Morakot portion of 2009 is less than one, and at nine of the twelve northern stations, the first post-Morakot values of $\tilde{a}$ are smaller than $\tilde{a}_{pre}$. At the remaining three stations, values of $\tilde{a}$ exceed

$\tilde{a}_{pre}$ by a factor of 1.3 (N3) to 2.1 (N2) in the post-Morakot portion of 2009, and by a factor of 2.9 (N11) in 2010—substantially smaller than the average post-Morakot increases in $\tilde{a}$ at the southern stations. Unlike at the southern stations, $\tilde{a}$ does not change systematically over time after Morakot at the northern stations.

As described in Section 2.1, we split the 2009 water year into pre-Morakot and post-Morakot portions, which permitted the first few months after Morakot to be isolated from the pre-Morakot portion of the water year. To identify these time periods

visually in Figure 4, the pre-Morakot portion of 2009 is plotted as an open circle and the post-Morakot portion of 2009 is plotted on the vertical line marking the time of Morakot. Splitting 2009 like this grouped most of the typhoon season into the post-Morakot portion of the year and excluded most of the typhoon season from the pre-Morakot portion of the year. Could this account for the observation that $\tilde{a}$ is higher during post-Morakot 2009 than it was before Morakot at some stations? We consider this unlikely. At many of the stations, the value of $\tilde{a}$ in post-Morakot 2009 is within 0.1 log units of that in 2010—

much closer than it is to the average value of $\tilde{a}$ in the years leading up to Morakot (e.g., at stations S1-S4 and S6-S7; Figure 4). If the high $\tilde{a}$ values in post-Morakot 2009 were only a result of splitting the year in two, then, all else equal, $\tilde{a}$ at the southern stations in 2010 ought to have been substantially lower, closer to the average $\tilde{a}$ value before Morakot.




**Figure 4.** Annual estimates of rating curve parameters $\tilde{a}$ and $b$ at the southern stations (S1-S12), where Morakot-induced landsliding was common, and the northern stations (N1-N12), where it was not. Vertical lines indicate timing of Morakot. At eleven of the twelve southern stations, $\tilde{a}$ increases immediately after Morakot relative to its pre-Morakot 2009 value (open circles) and then declines. $b$ does not respond systematically among the southern stations. By contrast, $\tilde{a}$ and $b$ at the northern stations show smaller responses to Morakot, confirming that basins with minimal landsliding experienced smaller changes in rating curves. At most stations, the timing of maximum $Q$ during this period (white stars) does not coincide with large changes in $\tilde{a}$ and $b$, implying that Morakot-induced changes in $\tilde{a}$ and $b$ were likely driven by changes in landslide-derived sediment supply, not changes in $Q$.





Unlike values of $\tilde{a}$, values of $b$ across the southern stations did not change systematically immediately after Morakot (Figure 4). Instead, $b$ increased at some stations and decreased at others. At the southern stations, the average $b$ values in the post-

Morakot portion of 2009 and in 2010 were 0.84 and 0.83, respectively, both slightly higher than the average $b_{pre}$ of 0.81 before Morakot. In the post-Morakot portion of 2009, $b$ was lower than $b_{pre}$ at three of the eight southern stations with measurements. In 2010, $b$ was lower than $b_{pre}$ at five of the twelve southern stations (Supplementary Tables S2, S4). Thus, $b$ decreased after Morakot at less than half of the southern stations in both 2009 and 2010, despite widespread landsliding in these basins.

The largest decreases in $b$ did not occur until 2011, when $b$ was smaller than $b_{pre}$ at ten of the twelve southern stations. In

2011, the average difference between $b$ and $b_{pre}$ at the southern stations was -0.38 and as large as -1.40 (station S1). At some of these stations, values of $b$ remained lower than $b_{pre}$ from 2011 through 2020 (S1, S12), while and at other stations, $b$ returned to approximately $b_{pre}$ in four to seven years (S2, S4, S5, S11). Thus, persistently lower values of $b$ appeared in several basins with intense landsliding, but not until the second year after Morakot.

By comparison, across the northern focus stations, where Morakot-induced landsliding was minimal, the average value of

$b$ dropped from 0.59 before Morakot to 0.29 in 2010—a greater decrease than the southern stations experienced during the same time frame, on average. Together, these observations show that some of the stations exhibited a post-Morakot decline in $b$ similar to that documented by Huang and Montgomery (2013), while others did not.

### 3.2 Suspended sediment discharge $Q_s$

At eleven of the twelve southern stations (all but S8), suspended sediment discharge increased within one year of Morakot

relative to average values before Morakot (Figure 5). At nine of these stations, suspended sediment discharge dropped within a year or two after Morakot, and at the remaining two stations (S9 and S10), suspended sediment discharge remained high for 3-4 years. At stations S1-S7, sediment discharge peaked in the post-Morakot portion of water year 2009 (August 6 to October 31, 2009). At stations S9-S12, sediment discharge peaked in 2010. For stations S9-S11, we cannot rule out the possibility that $Q_s$ peaked in the post-Morakot portion of water year 2009, given the absence of measurements at these stations during this

time.

The post-Morakot peaks in $Q_s$ were large. For the stations with peaks in $Q_s$ in the post-Morakot portion of 2009 (S1-S7), $Q_s$ during this time was 2.1-222 times higher than the median $Q_s$ before Morakot. For the stations with peaks in $Q_s$ in 2010 (S9-S12), $Q_s$ during this time was 3.4-84 times higher than the median $Q_s$ before Morakot. The rapid drop-off in $Q_s$ after Morakot was largely due to the influence of discharge $Q$ on estimates of $Q_s$ (Equation 1). For example, at site S1, the maximum daily

discharge in water year 2010 was only 18% of what it was during Morakot, and in 2011 it was only 13%. The total sediment discharged at S1 in all of water years 2010 and 2011 was only 4% and 1%, respectively, of the total during the last three months of water year 2009, which included Morakot. All but one of the other southern stations show similarly smaller floods in 2010 and 2011 than in 2009. With the exception of station S12, the annual sediment discharge at all stations in both 2010 and 2011 was less than 25% of the sediment discharge in August-October 2009.



**Figure 5.** Suspended sediment discharge from 1991 to 2020 at the southern (left) and northern (right) gauging stations (Figure 1). Vertical line in 2009 shows the time of Typhoon Morakot. The open circle in 2009 represents the pre-Morakot portion of the water year, and the filled circle on the Morakot line represents the post-Morakot portion of the water year. At most of the southern stations, suspended sediment discharge increased greatly during Morakot, then tapered off shortly afterwards. By contrast, at most of the northern stations, suspended sediment discharge did not change during Morakot.

In Figure 5, each water year is indicated with a filled circle except the pre-Morakot portion of the 2009 water year, which is indicated with an open circle. At most stations, estimates of $Q_s$ in the pre-Morakot portion of the 2009 water year are lower than those before and after it. This is largely because this portion of the water year does not include some of the typhoon season, which tends to have more large precipitation events. At most of our study stations, river discharge $Q$ is smaller during this portion of the water year than it is during typhoon season, which results in a smaller $Q_s$ (Equation 1).





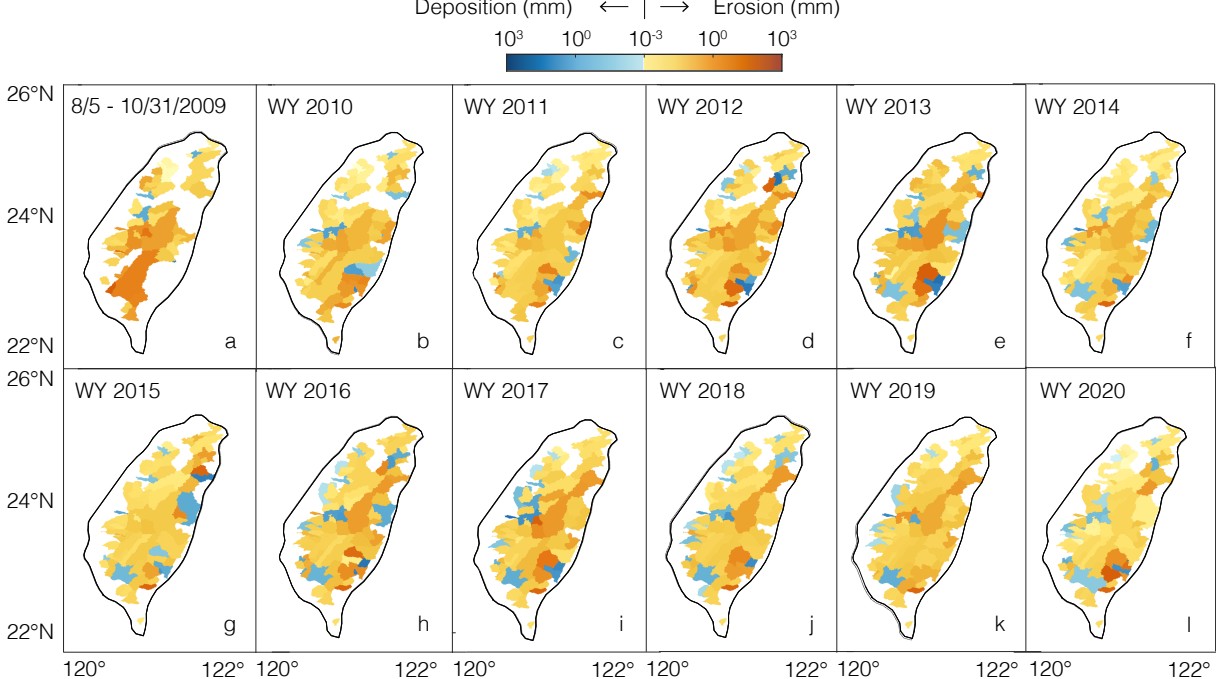

**Figure 6.** Basin-averaged erosion (yellow-red) and deposition (blue) in the first three months after Morakot (panel a) and each water year (November 1 – October 31) after that until 2020 (b-l), calculated with Equations 4-5. This shows that most of Taiwan was dominated by net erosion most years after Morakot, and that in many places the fastest erosion happened in the first few months after Morakot.

We used the estimates of $Q_s$ at all 87 stations to calculate basin-averaged erosion rate $E$ (Equation 4) each year after Morakot. Figure 6 and shows that $E$ varies greatly in time and space across Taiwan. In some small catchments in a given year, $E$ was exceptionally rapid. For example, the sediment discharge from the small basin above station 1660 H010 on the Erhjen River in the post-Morakot portion of 2009 is equivalent to $> 10^3$ mm of basin-averaged erosion. In other nested catchments, high $Q_s$ values at an upstream station are paired with low $Q_s$ values at a downstream station on the same river in the same

year, implying negative values of $E$ in Equation 5 and hence net deposition in the downstream portion of the basin that year. For example, the Beinan River basin, which contains stations S8-S10, shows an initial decline in erosion rates from 2010 to 2011. Beginning in 2011, erosion rates in the upstream portions of this basin increase dramatically and continue into 2013. Net deposition accumulates downstream, after which the basin returns to nearly uniform net erosion. Similarly, erosion rates in the downstream portion of the Zhuoshui basin (which contains sites S5-S7) decrease dramatically starting in 2010, resulting in net

deposition downstream for the remainder of the years analyzed. Together, the panels in Figure 6 show that most of the island is dominated by net erosion most years, and that the fastest erosion occurred in the first few months after Morakot in many regions.



## 3.3 Gain in suspended sediment discharge

Figure 7 shows that, at the southern stations, $Q_s$ was larger than $Q_{s,hypothetical}$ at some stations in some years and smaller
than it at other stations and in other years. For example, at station S1, $Q_s$ was 16 times larger than $Q_{s,hypothetical}$ in the post-
Morakot portion of 2009 and 4 times larger than it in 2010. At station S6, by contrast, $Q_s$ was half as large as $Q_{s,hypothetical}$
in the post-Morakot portion of 2009 and two times larger than it in 2010.

On average, $Q_s$ tended to be substantially larger than $Q_{s,hypothetical}$ across the southern stations for less than two years after
Morakot (Figure 7). $Q_s$ exceeded $Q_{s,hypothetical}$ at nine of the twelve southern stations in the post-Morakot portion of the 2009
water year (all but S5-S7), nine of the twelve stations in 2010 (all but S3, S4, and S7), and ten of the twelve stations in 2011
(all but S4 and S7). On average across the southern stations, $Q_s$ was 3.5 times higher than $Q_{s,hypothetical}$ in the post-Morakot
portion of 2009 and 9.1 times higher in 2010 (Figure 7).

From 2011 onward, however, $Q_s$ was only moderately larger than $Q_{s,hypothetical}$ at the southern stations, and then only
at some stations. Between 2012 and 2020, $Q_s$ exceeded $Q_{s,hypothetical}$ more than half the time at only 7 of the 12 southern
stations. On average, $Q_s$ was 0.8-2.6 times as large as $Q_{s,hypothetical}$ across the southern stations from 2012 to 2020.

By contrast, at the northern stations, $Q_s$ does not systematically exceed $Q_{s,hypothetical}$ at any time in the decade after
Morakot. In the post-Morakot portion of 2009, $Q_s/Q_{s,hypothetical}$ ratios are 13 and 4 at stations N2 and N3, respectively, but
are not significantly above 1 at any of the other northern stations. This reflects the absence of a strong suspended sediment
response to Morakot at the northern stations, even in the few months immediately after Morakot.





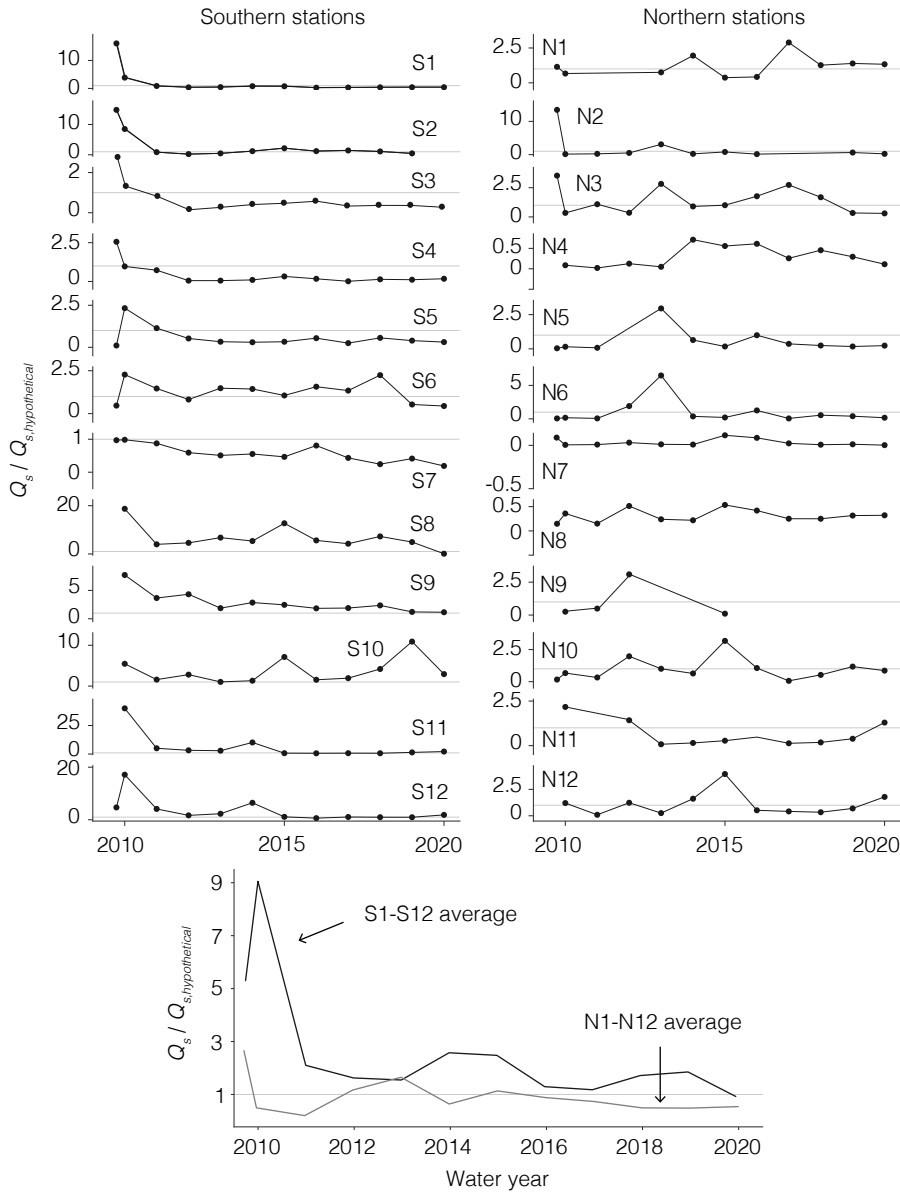

**Figure 7.** $Q_s/Q_{s,hypothetical}$ after Morakot at the focus gauging stations (Figure 1). Horizontal gray lines indicate $Q_s/Q_{s,hypothetical} = 1$. Note different y-axis extents in the individual station plots. At all but one of the southern focus stations, $Q_s$ was elevated above $Q_{s,hypothetical}$ at some point in the first two years after Morakot. By contrast, at seven the twelve northern stations, $Q_s$ does not exceed $Q_{s,hypothetical}$ at any time in the first two years after Morakot. The bottom panel shows the average $Q_s/Q_{s,hypothetical}$ ratios across the southern and northern focus stations. This shows that $Q_s$ at the southern stations was, on average, substantially higher than $Q_{s,hypothetical}$ in the first two years after Morakot, but not at the northern stations.





How much extra sediment was mobilized in the study rivers in the decade after Morakot? To calculate this, we define a
new term, $\Delta Q_s$, as the difference between $Q_s$ and $Q_{s,hypothetical}$. We integrated $\Delta Q_s$ over the time from Morakot through
2020 to obtain the excess mass of suspended sediment discharged at each station over this decade. To facilitate comparisons
to landslide intensity, we divided $\Delta Q_s$ by drainage area ($A$) and bedrock density $\rho_r$ and termed this the excess sediment yield
(Figure 8) to ensure that this and $I_L$ share the same dimensions of volume per area.

Figure 8 plots the excess sediment yield against $I_L$. The slope of the best-fit log-log regression is $0.394 \pm 0.110$, showing
that the excess sediment yield tended to be larger in basins with more intense landsliding. The excess sediment yield is larger
than $I_L$ for 68% of the stations (data points above the 1:1 line in Figure 8). These are generally restricted to stations with low
landslide intensity, with none at $I_L > 10^5$ m$^3$ km$^{-2}$. This implies that landslide-derived material was not the source of all of
the excess suspended sediment discharged from these rivers in the decade after Morakot. Meanwhile, the excess sediment yield
was smaller than $I_L$ for the remaining 32% of the stations, all of which were in basins with high landslide intensity ($I_L > 10^4$
m$^3$ km$^{-2}$). In these rivers, by contrast, the volume of landslide-mobilized material was large enough to have supplied all the
excess sediment yield.

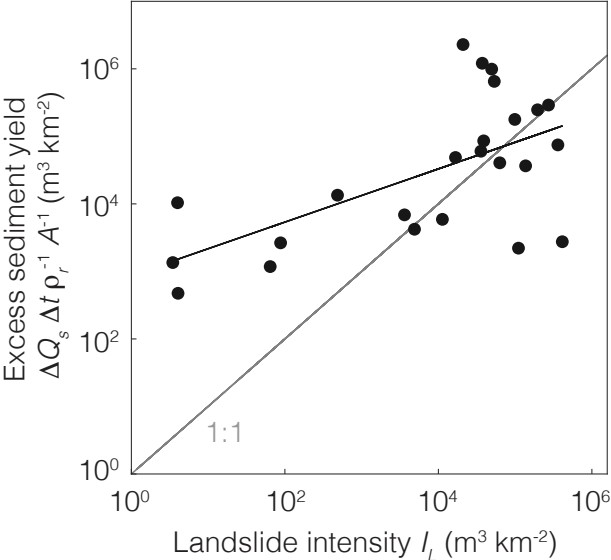

**Figure 8.** Excess suspended sediment yield ($\Delta Q_s = Q_s - Q_{s,hypothetical}$) per unit drainage area $A$ integrated over the time $\Delta t$ from Morakot
through 2020 as a function of landslide intensity $I_L$. Here, values of $Q_s$ have been converted from mass fluxes to volume fluxes by dividing
by a bedrock density $\rho_r$ of 2700 kg m$^{-3}$ to facilitate comparison to the units of $I_L$ we use elsewhere in this study (m$^3$ km$^{-2}$). The log-log
regression through these data (black line) has a slope of $0.394 \pm 0.110$, implying that excess sediment yield tended to be larger in basins
with more intense Morakot-induced landsliding.





## 4   Discussion

### 4.1   Duration of suspended sediment responses to Morakot

Many of the southern focus stations display a gradual decay in $\tilde{a}$ over time after Morakot (Figure 9). In Figure 9, we fit the post-Morakot decay of $\tilde{a}$ to a linear regression of $\ln(\tilde{a})$ against time, consistent with exponential decay. The mean and standard error of the regression slope quantify the rate of decay of $\tilde{a}$ and its uncertainty, and are listed in the panels of Figure 9. These regressions capture the reduction in $\tilde{a}$ well at 10 of the 12 southern stations, where the mean regression slope is larger than the standard error of the slope by an average factor of 4.5 (range 2.4-8.7). At the two remaining stations (S8 and S10), the

regression slope is zero within uncertainty, implying no resolvable change in $\tilde{a}$ (Figure 9).

By contrast, regressions of $\ln(\tilde{a})$ over time fit the data poorly at the northern stations. Regression slopes are indistinguishable from zero within one standard error uncertainty at six of the twelve stations (N4-N6, N8, N11, N12), positive at three stations (N1, N9, N10), and negative at the remaining three stations (N2, N3, N7; Figure 9). At none of the three stations with negative regression slopes does the mean regression slope exceed two times the regression slope's standard error. This shows that post-

Morakot trends in $\tilde{a}$ did not vary systematically across the northern stations, and that at the few stations at which $\tilde{a}$ did decline, the decline is less clear than it is at the southern stations.







**Figure 9.** Solid lines are linear regressions of the rating curve parameter $\ln(\tilde{a})$ (black dots) with respect to time. Numbers in each panel are mean ± standard error of the slope of the regression line. Ten of the twelve southern stations show a decreasing trend with time. In contrast, at the northern stations, regressions of $\ln(\tilde{a})$ against time have a range of positive and negative slopes with larger uncertainties. This suggests that the effects of Morakot decayed over time at most of the southern stations but not the northern stations.





How long did the Morakot-induced perturbations to $\tilde{a}$ last? To answer this question, we introduce the notation $\tau_{\tilde{a}}$ to denote the characteristic response time of $\tilde{a}$. In Figure 9, the linear regressions of $\ln(\tilde{a})$ vs. time describe exponential decay of $\tilde{a}$, which makes it convenient to define $\tau_{\tilde{a}}$ as the negative reciprocal of the regression slope (i.e., $\tilde{a}(t) \propto e^{-t/\tau_{\tilde{a}}}$). Stations with rapid

declines in $\tilde{a}$ after Morakot have large regression slopes and hence small values of $\tau_{\tilde{a}}$ (short response times), while stations with slow declines in $\tilde{a}$ have small regression slopes and large values of $\tau_{\tilde{a}}$ (long response times). Under this definition, $\tau_{\tilde{a}}$ ranges from 4.4 (+2.5/-1.2) years to 9.0 (+1.7/-1.3) years at the ten southern stations that have declining values of $\tilde{a}$ after Morakot. A property of exponential decay is a return to values within 5% of background after roughly three characteristic decay times, so a return to near-background values at these stations would occur after two to three decades. At the remaining

two stations (S8 and S10), the regression slopes are indistinguishable from zero within one standard error, implying values of $\tau_{\tilde{a}}$ indistinguishable from infinity. Uncertainties on $\tau_{\tilde{a}}$ are calculated as the negative reciprocals of the one standard error uncertainty bounds on the regression slopes in Figure 9 (Table S2).

To quantify the post-Morakot changes in $\tilde{a}$ in rivers across Taiwan, we calculated regression slopes of $\ln(\tilde{a})$ vs. time for all stations with landslide intensities greater than 1 m$^3$ km$^{-2}$ and with sufficient observations to compute post-Morakot trends

(n = 40; Table S2). Twenty-six of the 40 stations have negative regression slopes, indicating a post-Morakot decline in $\tilde{a}$. The remaining 14 stations have positive regression slopes, indicating a post-Morakot increase in $\tilde{a}$. Figure 10a reveals that, in the 25 basins with relatively high landslide intensities ($I_L > 1000$ m$^3$ km$^{-2}$), regression slopes tend to be more negative at higher landslide intensity. Moreover, among these 25 high-$I_L$ basins, only three have positive regression slopes, while 16 have negative regression slopes and the remaining six have regression slopes indistinguishable from zero. This reflects both the

tendency of $a$ to decay more rapidly in high-$I_L$ basins as well as the variability in decay rates of $a$ among basins, such as that exhibited by the difference between stations S8 and S10 (which have regression slopes indistinguishable from zero; Figure 9) and the other ten southern focus stations, which all have negative regression slopes. By contrast, in the 15 basins with relatively low landslide intensities ($I_L < 1000$ m$^3$ km$^{-2}$), regression slopes tend to be close to zero or slightly positive and uncorrelated with landslide intensity. Only four of these 15 low-$I_L$ basins have negative regression slopes, while the remaining eleven have

regression slopes that are positive or indistinguishable from zero.

We calculated $\tau_{\tilde{a}}$ for all stations with negative regression slopes to obtain characteristic decay times for the 26 stations at which $\tilde{a}$ declines after Morakot. How does the recovery time of $\tilde{a}$ depend on the intensity of landsliding? Figure 10b shows that $\tau_{\tilde{a}}$ tends to be shorter at larger values of $I_L$. For instance, the mean $\tau_{\tilde{a}}$ is 6 years in the basins with the largest landslide intensities ($I_L > 10^5$ m$^3$ km$^{-2}$) and 37 years at landslide intensities smaller than that. This indicates faster fractional responses

of $\tilde{a}$ in basins that were hit harder by landsliding.





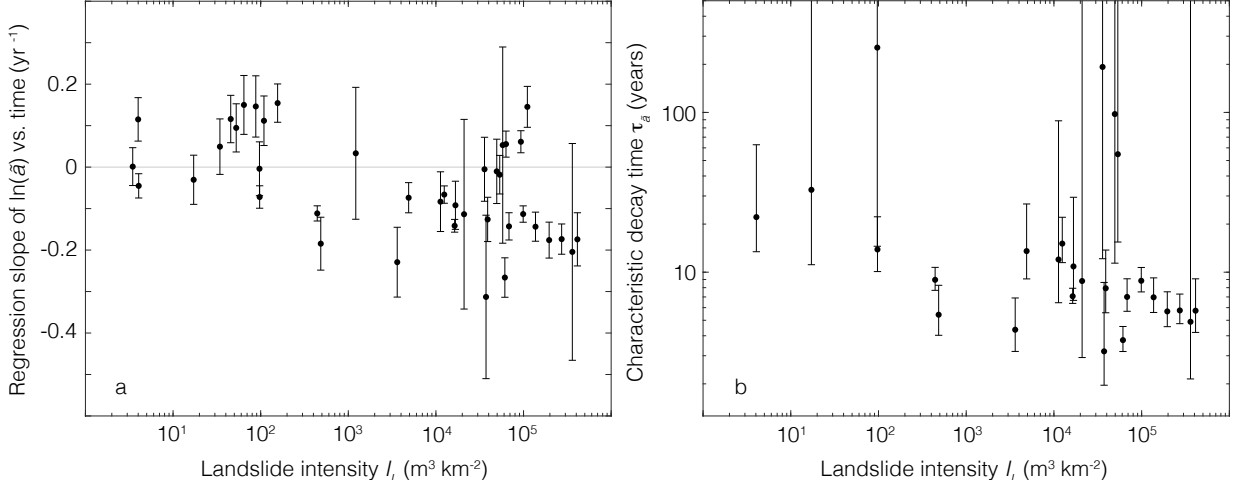

**Figure 10. a.** Means and standard errors of the slopes of the $\log(\tilde{a})$ vs. time regressions (Figure 9) for the 40 basins with non-zero landslide intensity. Twenty-six of the 40 regression slopes are negative, indicating decay of $\tilde{a}$ over time, and the remaining 14 are positive, indicating growth. **b.** Characteristic decay times of $\tilde{a}$ for the stations with negative regression slopes, calculated as the negative reciprocal of values in Figure 10a. Values of $\tau_{\tilde{a}}$ tend to be smaller at higher $I_L$ values, implying that the elevated efficiency of suspended sediment transport after Morakot decayed more quickly in basins hit harder by landslides.

Together, Figures 9-10 suggest that sediment transport was more efficient (i.e., $\tilde{a}$ was higher) after Typhoon Morakot than it was before it, but that this elevated efficiency should only persist for a geologically short time (no longer than a few decades) and only in basins with abundant landslides. This implies that large landslide deposits in these rivers should persist for long times. This is consistent with Chen et al. (2020), who observed that [10]Be concentrations in stream sediment in some rivers in

Taiwan were lower in 2016 than they were before Morakot, which can be interpreted as indicating a Morakot-derived pulse of sediment to the river network in 2009 that had not yet been transported away by 2016. The results are also consistent with DeLisle et al. (2022), who observed channel sediment aggradation of tens of meters in the steep, upper reaches of catchments in southern Taiwan, and who suggested that this sediment may take several centuries to excavate in some channels. Such a protracted duration of landslide sediment export may reflect the large volume and coarse grain size of landslide-derived

sediment relative to the river's transport capacity, at least for basins with intense landsliding (Yanites et al., 2010; DeLisle et al., 2022; Marc et al., 2021).

## 4.2 Influence of landslide intensity on rating curves

How much did the intensity of landsliding affect the magnitude of the responses in the rating curve parameters? Here we examine the sensitivity of the Morakot-induced changes in the rating curve parameters to basin-averaged landslide intensity.

To facilitate this comparison, we define $\Delta\log(\tilde{a})$ as the change in $\log(\tilde{a})$ from before Morakot to a given time after it.

$$\Delta\log(\tilde{a}) = \log(a_{post}) - \log(a_{pre}) \tag{8}$$



Here $\tilde{a}_{post}$ is the value of $\tilde{a}$ in the post-Morakot period of interest and $\tilde{a}_{pre}$ is the average value of $\tilde{a}$ during the monitoring period before Morakot, as defined in Section 2.1. For example, if we denote $\tilde{a}_{2009PM}$ to be the value of $\tilde{a}$ in the post-Morakot portion of water year 2009, then $\Delta\log(\tilde{a}) = \log(\tilde{a}_{2009PM}) - \log(\tilde{a}_{pre})$. We define $\Delta b$ the same way in Equation 9.

$\quad \Delta b = b_{post} - b_{pre}$ (9)

Here $b_{post}$ is the value of $b$ in the post-Morakot period of interest and $b_{pre}$ is the average value of $b$ in the monitoring period before Morakot. Figure 11a plots $\Delta\log(\tilde{a})$ against $I_L$ for the post-Morakot portion of 2009 for all basins with nonzero $I_L$. Here, each data point represents a single gauging station. A linear regression through these data has a slope of $0.11 \pm 0.05$ log(ppm)/log(m$^3$ km$^{-2}$), indicating that, during the first few months after Morakot, values of $\tilde{a}$ tended to increase more in
basins that were hit harder by landsliding. Analogous figures for each year after Morakot are shown in Supplementary Figure S1.





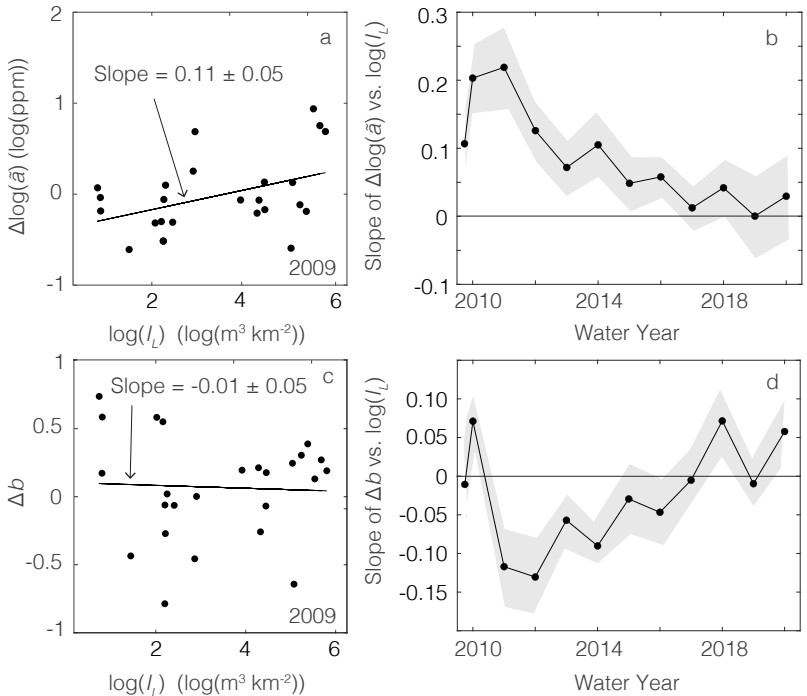

**Figure 11. a.** Sensitivity of the rating curve coefficient $\tilde{a}$ to the basin-averaged intensity of Morakot-induced landslides $I_L$. $\Delta\log(\tilde{a})$ is the difference in $\log(\tilde{a})$ between a given time (in panel a, the post-Morakot portion of 2009) and the pre-Morakot value $\tilde{a}_{pre}$. The regression line's positive slope shows that $\tilde{a}$ tended to increase more in basins that were harder hit by Morakot-induced landsliding. Analogous figures for other years after Morakot are shown in Supplementary Figure S1. **b.** Annual means (dots and line) $\pm$ standard errors (shaded region) of regression slopes of $\Delta\log(\tilde{a})$ against $\log(I_L)$ for each year after Morakot, computed as in panel a. The strength of the correlation between $\Delta\log(\tilde{a})$ and $I_L$ gradually decreased over time, implying that the sensitivity of $\tilde{a}$ to $I_L$ was no longer apparent by $\sim$7 years after Morakot. **c.** As in panel a, except showing the change in $b$ relative to the pre-Morakot value $b_{pre}$. The regression slope is indistinguishable from zero within error, indicating that $b$ was insensitive to the intensity of Morakot-induced landsliding during this time. Analogous figures for other years are shown in Supplementary Figure S2. **d.** Changes in $b$ were positively correlated with $I_L$ in 2010, then negatively correlated for several years in a row. This is consistent with a weak sensitivity of $b$ to the intensity of landsliding for four years after Morakot.

Figure 11b shows the strength of the relationship between $\Delta\log(\tilde{a})$ and basin-averaged landslide intensity over the decade after Morakot. The strength of the positive correlation between $\Delta\log(\tilde{a})$ and $I_L$ peaked at $0.22 \pm 0.06$ log(ppm)/log(m$^3$ km$^{-2}$) in 2011 and gradually grew weaker for several years after that. From 2017 onward, the mean regression slope of the correlation 410 was indistinguishable from zero within one standard error. Thus, the correlation between increases in $\tilde{a}$ and landslide intensity persisted for roughly six years after Morakot.

Figure 11c plots $\Delta b$ against $I_L$ for the post-Morakot portion of 2009. As in Figure 11a, each data point represents a single gauging station. A linear regression through these data has a slope of -0.01 $\pm$ 0.05 (log(m$^3$ km$^{-2}$))$^{-1}$, indicating that, during



the first few months after Morakot, values of $b$ were insensitive to the intensity of landsliding. Analogous figures for each year
after Morakot are shown in Supplementary Figure S2.

Figure 11d shows the strength of the relationship between $\Delta b$ and basin-averaged landslide intensity over the decade after
Morakot. This relationship was positive in the post-Morakot portion of water year 2009 and in 2010, indicating a brief period
in which $b$ values were positively correlated with landslide intensity. Then, in 2011, they became negatively correlated, and
remained so for several years. The strength of the negative correlation peaked at -0.13 ± 0.05 $(\log(\mathrm{m}^3\ \mathrm{km}^{-2}))^{-1}$ in 2012 and
gradually grew weaker for several years after that. By 2015, the mean regression slope was indistinguishable from zero within
one standard error. Thus, the negative correlation between $\Delta b$ and landslide intensity persisted for roughly four years from
2011 to 2014.

Together, the results in Figure 11 imply that the sensitivity of rating curve parameters to landslide intensity is resolvable
in this group of gauging stations for 4-6 years after Morakot, and slightly longer for $\log(\tilde{a})$ than $b$. This is comparable to the
average duration of elevated values of $\log(\tilde{a})$ and lowered values of $b$ (Figure 4). Beyond that time, there is no discernible
influence of Morakot-induced landslide intensity on changes in $\log(\tilde{a})$ and $b$. This is consistent with a persistent, decaying
influence of sediment supply changes on the rating curve parameters over roughly half a decade.

### 4.3    Potential drivers of post-Morakot variations in rating curves

As described in Section 3.1, estimates of $\tilde{a}$ increased immediately after Morakot at 11 of the 12 southern stations (all but
S6). What was responsible for this systematic increase? Huang and Montgomery (2013) noted that a shift from coarser to
finer suspended sediment would generate a decrease in $b$ at the same time as an increase in $\tilde{a}$, which they used to explain the
coincident increase in $\tilde{a}$ and decrease in $b$ at 15 of the 19 gauging stations they analyzed in the first two years after Morakot
(2009-2011). This interpretation was supported by bed grain size measurements made by Taiwan's Water Resources Agency
at one station in the Beinan River, which revealed a reduction in median grain size after Morakot (Huang and Montgomery,
435    2013).

We are unaware of pre-Morakot and post-Morakot grain size measurements at other stations, but if Morakot induced a
reduction in grain size at all the study rivers, then this would explain the systematic increase in $\tilde{a}$ immediately after Morakot. It
also implies that $b$ should have decreased at the same time that $\tilde{a}$ increased. Contrary to this expectation, however, $b$ was lower
than $b_{pre}$ in 2009 and 2010 at less than half of the stations at which $\tilde{a}$ exceeded $\tilde{a}_{pre}$. In the post-Morakot portion of 2009, $b$
was lower than $b_{pre}$ at only two of the seven southern stations with sufficient measurements (S5 and S7; Tables S2 and S4). At
the five other stations (S1-S4 and S12), $b$ increased during this time, rather than decreasing. In 2010, $b$ was lower than $b_{pre}$ at
five of the eleven stations at which $\tilde{a}$ exceeded $\tilde{a}_{pre}$ (S1, S2, S5, S7, and S10), but $b$ was higher than $b_{pre}$ at the other six. This
brief period of elevated $b$ values in 2010 was followed by 3-6 years in which $b$ values were, on average, lower than pre-Morakot
values. Thus, across the southern study basins, where landsliding was prevalent, the systematic reduction in $b$ values occurred
in 2011, roughly 1.3 years after Morakot and the increase in $\tilde{a}$ values. To the extent that the reduction in $b$ reflects a shift from
threshold bed to live bed sediment transport (Huang and Montgomery, 2013), this suggests a brief period of adjustment toward
these conditions after Morakot.



Could other events, like additional typhoons, have affected estimates of the rating curve parameters after Morakot? At many of the southern stations, there is considerable interannual variation in $\tilde{a}$ around the downward trend after Morakot, which is reflected in the uncertainties on the regression slopes in Figure 9. Typhoon Fanapi, for example, brought intense rainfall to southern Taiwan in September 2010, and the estimates of $\tilde{a}$ at most of the southern focus stations in 2010 lie above the regression slopes in Figure 9. Could Fanapi have introduced more landslide-derived sediment to the study rivers than Morakot did?

The fluvial water discharge measurements suggest this is unlikely. At the Beinan River station that is farthest downstream (WRA station 1730 H043; Table S1), the maximum avg. daily discharge during Typhoon Fanapi was 2,800 m$^3$ s$^{-1}$, roughly 15% of the maximum discharge during Morakot (nearly 15,000 m$^3$ s$^{-1}$). To the extent that landslide occurrence is correlated with river discharge, this suggests that Fanapi may have contributed to elevated values of $\tilde{a}$ in 2010, but that the responses in $\tilde{a}$ to Fanapi were likely smaller than the responses to Morakot. It is possible, however, that in addition to contributing landslide sediment after Morakot, runoff from Fanapi (the first major typhoon of the 2010 season to hit the Beinan basin) may have introduced additional sediment from Morakot-induced landslides to the fluvial network, in the same way that Typhoon Toraji did in 2001 after the 1999 Chi-Chi earthquake (e.g., Dadson et al. 2003). This is supported by the study of Hung et al. (2018), which showed that the destabilizing effects from Morakot may have contributed to increased landslide intensity for ∼ 5 years after the typhoon. Teng et al. (2020) also demonstrate through numerical modeling that reactivated, old landslide material can influence sediment transport after a large landsliding event.

A counterexample is Typhoon Soulik, which was the largest typhoon to hit Taiwan in 2013, the most active typhoon season in Taiwan since 2004. Soulik produced heavy rainfall on both the north and south sides of the island, peaking at > 600 mm on July 13 (Wu et al., 2018). This coincided with a small peak in $\tilde{a}$ in 2013 at some stations on the northwest side of the island (N1-N6), but relatively small responses at the remaining northern stations and most southern stations (Figures 4, 9). Among the southern stations, only S3 shows a local maximum in $\tilde{a}$ during 2013, when $\tilde{a}$ was 68% higher than in 2012. This suggests that Soulik had a relatively small effect on suspended sediment transport in most of the study rivers, unlike Morakot, underlining the fact that the Morakot-induced increases in sediment fluxes were likely due to the combined effects of intense landsliding and flooding, rather than flooding alone.

A final potential driver of post-Morakot changes in $a$ and $b$ relates to the channel itself. If the channel cross-sectional geometry changed during Morakot (e.g., through widening or deepening), then a given $Q$ could generate a different basal shear stress after Morakot than before it, which in turn could generate a different relationship between $C$ and $Q$ and hence different values of $a$ and $b$. Identifying any such effects are beyond the scope of this study, but may be useful in future studies to help interpret sediment discharge estimates at individual stations.

## 5 Conclusions

The primary contribution of this study is a new assessment of the effects of Typhoon Morakot on fluvial suspended sediment loads over an 11-year period after Morakot in 87 rivers around Taiwan. The most striking signal is a peak in the rating curve

coefficient $\tilde{a}$ within one year of Morakot, with larger fractional increases in $\tilde{a}$ in basins with more intense landsliding. This was followed by a decline in $\tilde{a}$ with an exponential characteristic decay time of 3-255 years, with shorter (sub-decadal) decay times in basins with more intense landsliding. By contrast, the rating curve exponent $b$ did not drop systematically until two years after Morakot, even in basins with abundant landsliding. The post-Morakot increases in $\tilde{a}$ and decreases in $b$ tended to be

larger in basins with more intense landsliding, but this sensitivity to landslide intensity decayed away within 4-7 years. These changes resulted in a positive correlation between excess suspended sediment yield and landslide intensity integrated over the decade after Morakot.

Together, these observations are consistent with an influence of landsliding on suspended sediment transport efficiency that was large immediately after Morakot and then diminished rapidly. This implies that the influence of Morakot-induced

landsliding on suspended sediment concentrations substantially declined within the first decade after the typhoon and that its influence will disappear entirely within a few decades. To the extent that these results are applicable to other mountainous rivers, this suggests that rivers may be able to move landslide-derived sediment more efficiently for only a few years to decades after landslide events. Thus, although landslide deposits in river valleys may persist for centuries, elevated suspended sediment concentrations may only last a short fraction of that time.

*Data availability.*  Taiwan river data is available from the WRA hydrological yearbook (Water Resources Agency, 2020):

https://gweb.wra.gov.tw/wrhygis/

*Author contributions.*  GR performed data analysis, wrote the code and manuscript and helped conceive the study. KF conceived the study, contributed ideas, wrote the manuscript, and performed data analysis. OM provided landslide data, contributed ideas for data analysis, and wrote the manuscript.

*Competing interests.*  No competing interests.

*Acknowledgements.*  We thank Bruce Shyu and Meng-Long Hsieh for meeting with us and providing helpful discussion. We thank Bruce Wilkinson for his edits on the manuscript. This study was made possible by NASA grant 80NSSC17K0098, "Cascading Hazards" which provided funding for Ferrier and Ruetenik.



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
