# Peer review of "Decadal-scale decay of landslide-derived fluvial suspended sediment after Typhoon Morakot"

_EGUsphere, 2023_

## Referee Comment (RC1)

Review of *Decadal-scale decay of landslide-derived fluvial suspended sediment after Typhoon Morakot* submitted to Earth Surface Dynamics

Ruetenik and co-authors present a very interesting assessment of multi-year suspended sediment flux data from around Taiwan and assess how suspended sediment discharges and its relationship to hydrology is affected by typhoon Morakot. They compare these data with an inventory of landslides that were triggered by the typhoon and make inferences about the timescales of sediment evacuation from landslides after extreme events.

I found this paper to be a very well written and presented contribution that just leaves me with a few suggestions and comments that I invite the authors to consider.

In particular, I wonder about the conclusion that landslides are driving the change in rating curve parameters. You show that many of the catchments have excess sediment yield that is above the landslide yield (by orders of magnitude) – e.g. Figure 8. Does that mean that you are measuring activation of non-landslide parts of the landscape during typhoon Morakot? If you are measuring a substantial proportion of sediment discharge from non-landslide parts of the landscapes, would it be possible that the change in the rating-curve parameters is driven by non-landslide parts of the landscapes just as much as the landslide parts? The correlation with landslide intensity could then be due to a co-variation of Typhoon Morakot intensity with landslide intensity. If you plotted Figure 8 and Figure 10 with, for example, rainfall intensity from Morakot instead of landslide intensity, would you find a similar result?

I suggest to give a bit more space to the impact and the implications of looking specifically at suspended sediment transport. In particular, I wonder about the conclusion that the "periods of elevated sediment transport efficiency after landslides should persist from years to decades" (L26-27). Isn't it possible that bedload transport in larger floods will be elevated for many more years? In regards of the discussion of previously measured sediment evacuation times in L36 – 37: As far as I understand, at least some of the references that are cited look specifically at bedload transport (Croissant et al., 2017; Yanites et al., 2010), so the times of export may be quite different. Finally, on a minor point, when comparing suspended sediment yield and landslide yield, in Figure 8 and in the associated discussion, I presume the landslide yield includes all grain sizes, so should be a bit lower when compared with suspended sediment yields, right?

I wonder about whether north-south changes in lithology could underly some of the observed north-south trends. My intuition is that this effect should be minor and it is also hard to test for, but maybe worth adding a line about lithology somewhere.

**Line comments**

L8-9: I can imagine that some readers do not have an intuition of what changes in the coefficient and exponent of the rating curve mean in terms of process (or maybe they do not know what these parameters represent). If there is a way to describe the changes in words and/or define the parameters, that might help. (e.g. instead of saying that the coefficient was a

factor of 5 higher, you could say that the suspended sediment transport for a given discharge was a factor of 5 higher etc.)

L17: "Shortly after […]". Sounds like a repetition of information from previous sentences – may be streamlined.

L37: With a brief look at the cited reference, I can only see estimates for evacuation times of 250 – 600 years, not thousands of years. Also, as mentioned above, these are, as far as I understand, estimated for bedload transport, not for suspended sediment transport as suggested in this sentence.

L110: If eight basins show no landsliding due to Morakot – why not use these eight and instead add the other basins to the group that have landslides?

L248 / Figure 4: As far as I understand from the definition in L182, the pre-Morakot values are averaged across the entire period pre Morakot. Here, you have another pre-Morakot value that is just the part of the year 2009 before Morakot (empty circle in Fig. 4) –A different designation for these different measurements would be clearer.

L296: That sentence doesn't work – maybe the "and" needs to go?

L337/Figure 8: By plotting the figure in log-space. You are discarding negative values. I wonder what'd happen to the correlation, if you plot it in linear space? Do the catchments with negative values fall on a similar trend?

L340/Figure9: Can you plot the pre Morakot $\tilde{a}$-values in the figure, for example as a horizontal/shaded underlay? That would be great to see if the values recover to the previous value or even overshoot.

L377 – 78: Given the high uncertainties in Fig 10b, I wonder what the likelihood is that there is no difference in the values at low and high landslide intensity. Would it be possible to add a statistical test here?

L384 – 386: My understanding of Chen et al. (2020) is that by 2016, the $^{10}$Be concentrations are basically not statistically distinguishable from before Morakot when considered across all catchments. Of course, there are some catchments that have lower $^{10}$Be than before Morakot but there are also some that have higher concentrations. This sentence suggests to me that the effect of Morakot is still clear in 2016.

L395: Here and between figures and the text, you switch between $log(\tilde{a})$ and $ln(\tilde{a})$. I presume that $log$ is the $\log_{10}$ and $ln$ is $\log_e$? Is there a reason for considering the natural logarithm and sometimes the base 10 logarithm?

L407/Figure 11: Given the poor fit in panel c, I wonder if the goodness of fit should somehow come into panels b and d? Also, the "strength of the relationship" sounds a bit like a goodness of fit criterion. Maybe you can find a different wording? For example, "sensitivity of $\Delta log(\tilde{a})$ to landslide intensity.

I hope that the comments are helpful, and I remain with best wishes to the authors and editor.

Aaron Bufe

**References**

Chen, C.-Y., Willett, S. D., West, A. J., Dadson, S., Hovius, N., Christl, M., and Shyu, J. B. H., 2020, The impact of storm-triggered landslides on sediment dynamics and catchment-wide denudation rates in the southern Central Range of Taiwan following the extreme rainfall event of Typhoon Morakot: Earth Surface Processes and Landforms, v. 45, no. 3, p. 548-564.

Croissant, T., Lague, D., Steer, P., and Davy, P., 2017, Rapid post-seismic landslide evacuation boosted by dynamic river width: Nature Geoscience, v. 10, no. 9, p. 680-684.

Yanites, B. J., Tucker, G. E., Mueller, K. J., and Chen, Y.-G., 2010, How rivers react to large earthquakes: Evidence from central Taiwan: Geology, v. 38, no. 7, p. 639-642.

---

## Referee Comment (RC2)

To the editorial team at *Earth Surface Dynamics*,

I have completed my review of the manuscript "egusphere-2023-1278: Decadal-scale decay of landslide-derived fluvial suspended sediment after Typhoon Morakot" by Ruetenik et al. [https://egusphere.copernicus.org/preprints/2023/egusphere-2023-1278/]. The manuscript uses decades-long river suspended sediment and discharge measurements from a series of stations around Taiwan to investigate the impact of the 2009 Typhoon Morakot its resultant landslides on river suspended sediment delivery over time. The authors show that among a subset of focus stations after Typhoon Morakot, rivers generally had large increases in the rating-curve parameter $\tilde{a}$ (meaning that more sediment is delivered for a given discharge) and smaller decreases in the rating-curve parameter $b$ (meaning that the amount of sediment delivered becomes less dependent on the magnitude of discharge), both compared to pre-Morakot averages. The peak for $\tilde{a}$ occurred within ~1 year, while the decrease in $b$ peaked after a couple of years, and both values decayed back to pre-Morakot levels, with the most-affected regions decaying within a couple of decades.

Overall, after a close review, I have found the manuscript to be well-written, clear, and with few typographical or grammatical errors. It is clear that the submission has been treated with care. The work addresses relevant scientific questions to the readership of ESurf, presents novel and reasonable results, and has a method that seems technically sound and reproducible. While the manuscript requires little in the way of major changes, I have some minor comments for the editor's and authors' consideration. I have also included line-level comments. While I think some of my comments bear addressing, others are merely suggestions or observations.

I thank the authors for a well-written manuscript. I have enjoyed reading it, and I hope that my comments prove helpful.

Best regards,

Harrison Martin
Postdoctoral Scholar Research Associate
Division of Geological and Planetary Sciences
Caltech

**Minor:**

- The fundamental data of the study are centered around 87 river measurement stations in Taiwan. At different points in the manuscript, either the entire dataset, a subset of 24 stations split into North and South halves, or other subsets/splits are measured and reported-upon. At times, I found myself unsure of which stations were being analyzed and reported upon. I have noted in the line-level comments for some specific spots where I was unsure. Overall, I'd encourage more clarity about which stations are used for which analyses, and justifications for why.
- Many results are focused around the 24 focus stations. I'd like to see something explaining why these 24 were selected, evidence that they are fairly representative of the whole dataset, how they were split into North and South, and quantitative presentations of their associated parameters. In particular, part of the results hinges implicitly on the idea that the 24 stations and associated basins are comparable to one another, and vary only in that the northern 12 experienced less precipitation and less landsliding than the south. A series

of box-and-whisker plots, or table values with standard deviations, should be able to easily clear this up either at the end of Methods or start of Results. This would also help with justifying the split between North and South focus stations, and allow the landslide intensity data to be presented earlier (as they currently do not arrive until the final paragraphs/figure of the results).

- Some of the changes, such as the year-over-year changes in rating-curve parameters and Qs observed immediately after Morakot, would be more impactful to the reader if they were presented alongside some context of the significance compared to year-over-year changes before Morakot. Currently they are compared to pre-Morakot mean or median values, which by nature are central values without any context about variability. It would be nice to know if the magnitude of year-over-year changes is larger than, or within the range of, previous non-Typhoon "noise".

- The regressions (and interpretations) between rating curve and suspended sediment changes are done against landsliding intensity, as per the paper title. They are not, to the best of my knowledge, done against precipitation intensity, which is another reasonable mechanism that could be driving changes and for which data are available. It would be enlightening to regress changes against both to see if landsliding is the mechanism driving the change, or if both are correlated because they are both driven by precipitation. This is particularly interesting because there are a few Northern focus stations that apparently received precipitation but not landsliding, and yet reacted more similarly to the Southern stations than to the other Northern stations. Additionally, for most stations, the volume of excess sediment transported by rivers post-Typhoon (via suspended sediment alone) exceeds the volume of landslided materials resulting from the Typhoon. This suggests that non-landslided materials might also have made up a substantial part of the response, and this should be acknowledged and discussed. Regressing changes against rainfall intensity should help with this discussion.

**Lines:**

8-10: This is a bit tricky since it's an abstract and necessarily without elaboration, but I'd suggest offering a brief introduction to the coefficients here. This can even take the form of a brief phrase in a sentence, along the lines of " [...] the discharge-normalized rating curve coefficient $\tilde{a}$ was higher [...] the first year after Morakot (2010), indicating heightened sediment concentrations for given discharges." And similarly for $b$, saying it indicates a greater dependence of sediment concentration on discharge, or an increased sensitivity.

10: This could easily just be me, but seeing "Morakot (2010)" made me think it was a citation.

15-16: I think that "Values of a˜ tended to decline faster in basins with more intense landsliding" from 15 is repetitive with "Shortly after Morakot, changes in $\tilde{a}$ and $b$ tended to be larger in basins with more intense landsliding"?

22-27: An effective summary and an interesting result. I appreciate the clarification that just because the increase in sediment transport disappears on a short timescale, that does not mean that the sediment deposited by landslides disappears on short timescales. Not sure that it needs to be in the manuscript, but it does make me wonder where sediment transport rates and sediment availability become detached in these sort of systems. Perhaps it winnows away all of the fine

material and becomes transport-limited over short timescales, or maybe much of the landslide material is at elevations inaccessible to the river?

37: To the best of my understanding, Yanites et al. (2010) dealt with bedload transport, not suspended sediment. Additionally, unless you're referring to a different finding, according to Figure 2D the longest timescale for removal of (bedload) material is ~600 years. I'd suggest a different citation for this idea.

44-47: Interesting that such a big event only tripled the average annual pre-landslide sediment load. My intuition was that the increase would have been more significant.

85: Based on the manuscript's abstract and introduction so far, I get the impression that the manuscript is specifically focused on analyzing fluvial suspended sediment fluxes, so it might be worth adding that to this line.

106-110: This method of splitting up the data between north and south is framed as distinguishing between areas with lower and higher precipitation/landslide intensity, respectively. That's probably a good representation and valid, especially looking at Figure 1. It could be worth quantifying and reporting this here anyways to make this point quantitatively by presenting, for example, average precipitations per group). There may also be other factors beyond precipitation that varied systematically from south to north; Yanites et al. (2018: DOI: 10.1002/esp.4353) focused only on the southern tip, but found or discussed systematic latitude trends in maximum elevation, slope, local relief, channel steepness, erosion/exhumation rates, and channel width.

Figure 1: I find it a bit challenging to identify all of the small red dots because of their size, having the same color as the large red dots, and sometimes being apparently overlain by other dots. Perhaps enlarging them and using a third color could help. Additionally, I could see some readers wishing the colored contours using a perceptually uniform colorbar so as to not artificially introduce breaks, as jet tends to do [https://colorcet.com/].

126-127: Here is a nice, concise introduction to the meaning of the parameters; some version of this also mentioned above would be helpful.

131: I came back to this line after reaching section 2.3 and feeling that landsliding intensity had not been emphasized earlier in the methods. While it's mentioned in the Introduction and shown in Figure 1, I think it's worth making sure that you clarify that you are measuring the influence of Morakot-induced landsliding, and not Morakot itself (ex. precipitation). This fits better with the title of the paper and introduction. I'd also look at mentions in section 2.2 and the caption of Figure 3 to ensure you properly emphasize that you are estimating the effects of landsliding (due to Typhoon Morakot) on suspended sediment discharges.

134: Consider removing "applying".

135: I think that readers who have not constructed rating curves may not understand what centering means in an intuitive sense. Is it akin to horizontally or vertically shifting values in some log-space? It looks like it might be normalization, or comparison of values as deviations from a central value. A few words would be appreciated, especially since it seems Cohn et al. (1992) do not provide much in the way of explanation themselves.

143: Ahh! It seems this line contains the graphical explanation I was looking for above.

151: To each year's C and Q data for each station, right?

161: "sediment load" -> "suspended sediment load"

166: Where/how did you get or measure A for each gauging station? It isn't mentioned as part of the WRA dataset on lines 95-96. I'm assuming a DEM and some routing toolkit, but it bears mentioning.

169-170: A good disclaimer. I think it would be worth a half-sentence about how far off you might expect the suspended sediment load to be from the total sediment load. Is suspended load 10% of the total load? 50%, 90%?

Eqn (5): Should E's subscript be 1 instead of 3, since it is associated with downstream gauging station 1?

179-180: You say here that you apply Eqn's 1-5 to each gauging station's measurements each year, to yield estimates of parameters at each gauging station. However, in sections 3.1 and the first two paragraphs of 3.2, it seems that you only present the results (and statistics) for estimates of $\tilde{a}$, $b$, and $Q_s$ for the 24 focus stations. Then, for the second half of 3.2, you say you calculated $Q_s$ for all 87 stations and use them to calculate and present $E$ for all all 87 stations. I'm unclear as to the selection criteria for the focus stations and why only 24 were presented and used for summary statistics, if it appears that values were calculated for all 87. Do we believe that the focus stations are representative of all stations?

188: The tense elsewhere has been past, but is present here. This may be intentional but it's worth giving a look-over to ensure consistency.

188-193: The first sentence is a good topic, but the rest of this paragraph seems a bit out-of-place. It might work if you add what's missing after the first sentence: a connection that says we are aiming to quantify the effects of Morakot by using changes in rating curves before and after a given storm. Then the following lines justify that it's reasonable to anticipate seeing these changes in rating curves; in fact, we should expect to.

194-215: This is a nice and clear explanation of why average Qs rates pre- and post-storm aren't sufficient to answer the research question. Only note is that you may be able to combine the final two sentences (213-215) into the paragraph above (208-212).

222-223: Marc et al. (2018) is cited twice in the same phrase.

222-224: "in which it is assumed" as used here applies to "corrections", and not the catalogue estimates themselves. Is this intentional? I'd also define c and p here; you can still mention the values later in the paragraph, though if you did not do the estimations yourself (as you say on line 228) you might not need to mention the explicit values of p and c here.

229: Insert "for each station"

232: I think section 3 should include a summary of the differences between the North and South sections in terms of the "input" conditions, to communicate to the reader that they are comparable in all manners (drainage areas, typical discharges) except for the treatment condition (precipitation and landsliding intensity). Much of the following results relies on them being distinct in terms of precipitation and landsliding intensity, and presumably not in other ways. As such, why not include a figure with a couple of box-and-whisker or violin plots showing that there is a quantitative difference between the two groups in terms of precipitation and landslide

intensity and how significant the difference is or isn't? That would make statements like the first sentence of Figure 4 caption easier to justify and write, since it will be in readers' heads already.

234-235: refer to Figure 4

242-244: Is there a figure showing this anywhere? Could be simple and useful to communicate the differences between the two zones. Either as two box-and-whisker plots, or with station # on the x-axis and $\tilde{a}/\tilde{a}_{pre}$ on the y.

252: This is entirely up to personal style, but if one wanted to avoid rhetorical or prompting questions and reserve them for research questions or prompting discussion sections, one way to re-write this could be "While this could account for the observation [...] at some stations, we consider this unlikely."

252-257: This is a good argument, but I'm not sure it's necessary unless you expect readers to bring it up unprompted. Could be shortened if you wanted, though I always respect including disclaimers. On the other hand, if it were very important to prove this, you could do a similar calendar split on each of the pre-Morakot years and see how common or uncommon such a within-year jump in $\tilde{a}$ is.

Figure 4 caption: The final sentence is a result, and is not shown in this figure, so I do not believe that this idea should first appear here unless it is accompanied by a reference to an associated supplemental figure. Is it shown or discussed elsewhere? Additionally, I'm not sure that the word "confirming" in the phrase "[...] confirming that basins with minimal landsliding experienced smaller changes in rating curves" is the most appropriate, as I'm not sure we've seen quantitative evidence yet that the northern and southern stations differ systematically in landsliding intensity beyond that shown visually in Figure 2. It almost looks like it might correlate similarly well to precipitation intensity.

Section 3.1: This might be personal taste as well, but I'd consider including a table of summary results/statistics between the north and south areas. The table could present many of the average pre-Morakot and post-Morkot 2009/2010 values for the different parameters of interest ($\tilde{a}$, $b$, $Q_s$). That would replace much of the writing, but I suppose that might make this section quite brief, and might not work for comparisons that rely on looking at values in 2011 (such as $b$ in the south), so might not be the effort.

258-270: I think it might be helpful to put the magnitude of these changes into context for readers who are not as familiar with Taiwan or the parameters. It was not as big of an issue with values of $\tilde{a}$, where changes were on a factor of ~2-8x. Here, all we can see is that the changes at the southern stations from pre-Morakot to post-Morakot 2009-2010 were much smaller than the changes from pre-Morakot to 2011, and the northern changes from pre-Morakot to post-Morakot 2010 are somewhere in the middle. What we lack is an understanding of the relative scale between the largest changes we see here compared and the sort of changes over comparable timescales that would be expected from normal annual variations without a typhoon. An easy way to do this is to compare the magnitude of these changes to, say, the standard deviation of the values pre-Morakot or the average magnitude of changes between years. Essentially, when we see an average year-over-year drop from 0.59 to 0.29, is that an exceptionally strong signal, or is that within the range of noise we could see in pre-Morakot measurements and thus indistinguishable from random chance? It should be possible to do some sort of basic statistical test to see which of the changes

assigned to Morakot are significant vs insignificant, given the pre-Morakot history in annual rating curve parameters.

281-283: Above, you compared post-Morakot values of rating curve parameters to the mean pre-Morakot values, but here you compare post-Morakot values of $Q_s$ to the median pre-Morakot values. Any particular reason for the different method?

283-284: I think I know what you mean, but saying "rapid drop-off in $Q_s$ after Morakot" almost sounds like it declined after Morakot. I think it'd be more accurate to say that the drop-offs occurred after the post-Morakot peaks. Additionally, here (as well as the caption for Figure 4) a reference is made to changes in annual discharge values. Are the timeseries of discharge, or even just annual averages per-station or for north vs south, shown anywhere?

287-279: I'm not sure if the "all but one" station mentioned in the first line is station S12 mentioned in the second line, or if these two lines are related to one another.

Figure 5: Interestingly, N1-3 have increases in suspended sediment discharge that look like those of the southern stations. I wondered if N1-3 are the south-most of the Northern stations. Based on Figure 2, it looks like these stations are southern, but not as much as N11/12. It also looks like these stations did experience elevated precipitation compared to the other Northern stations, but N2 and N3 experienced no landslides. This leads one to wonder if some of the increased $Q_s$ observed is due to non-landslide associated mechanisms that also correlate to precipitation. It would be interesting to regress $Q_s$ separately against landslide intensity and against precipitation.

Figure 6: How are these basins delineated? Is there a DEM involved? Is it from the WRA dataset?

295-307: I think that these results should be in a section of their own, and not part of "3.2 Suspended sediment discharge $Q_s$". Additionally, though it was mentioned in the methods, it bears repeating here that this is basin-averaged erosion or deposition rates associated with suspended sediment, specifically (as opposed to total actual erosion which presumably would be higher as it included bedload, etc.).

296: remove "and".

297: Is this station better-known as S# or N#? I suppose not, if it's not one of the focus stations.

298: How does this compare to "normal", non-Typhoon annual basin-average erosion rates for this type of setting?

Figure 7: Missing the word "of" in "By contrast, at seven of the twelve northern stations [...]"

325-337: I like this presentation, and I also think it's a reasonable result to find that basins that over-deliver sediment have lower landslide intensities than those that under-deliver. I suppose this suggests that the range of $Q_s$ amounts delivered by rivers exists over a narrower range than that of landslide intensities, and that the former is only moderately sensitive to the latter. Have you tried this same regression against rainfall intensity (which also has units of $L^3/L^2$)? That might give some insight as to where some of the non-suspended sediment is coming from. It's also interesting that in most cases, $\Delta Q_s$ is greater than the volume of landslided material considering the former doesn't include bedload transport, and I imagine a lot of the landslide material is coarser than would be expected from suspended sediment. This suggests to me that either landslides somehow "prime" landscapes to deliver excess sediment, or much of this excess suspended sediment you observe is not, in fact, material that was captured in the 2018 landslide

dataset. Do you have a few words about what the character of this material might be, or the process that mobilizes it?

Figure 8: The caption says what is shown by each point, but not what each point is. Is each one of the 24 focus stations?

Figure 9: I'd recommend adding a horizontal line showing the pre-Morakot value for reference. Also, are these the same data as are shown in Figure 4? If so, any particular reason/justification for using $\log(\tilde{a})$ there (and later in section 4.2) and $\ln(\tilde{a})$ now?

377-380: The question and content asked and described here are presented earlier in the paragraph above, on lines 368-373. I'd suggest rearranging or combining.

Figure 10: I am thinking about how to compare the values shown here in b) to the take-away from lines 365-359, where the minimum decay timescale was ~4 years and the maximum was ~ 9 years. It seems that many here are >9 years.

383-391: This argument is reasonable, and I think it makes sense. It could probably be validated by looking at modern satellite images. However, I am not sure how to rectify it with the data shown in Figure 8, which shows that many/most of the stations have already delivered more excess sediment (purely via suspended load) than the total landslided volume. The volume of delivery would presumably be even larger if bedload were included. I think this should be acknowledged and/or discussed somewhere.

482: I believe that this is the first time that the characteristic decay time of 3-255 years has been presented, at least in words. Should be presented before the conclusions section. I assume that this is the total range among all stations, and the 4-9 years presented before is among a subset of stations (focus, or high-landslide-intensity and also decaying, etc.).

483-484: Do you have any intuition or understanding of why rates of $\tilde{a}$ should respond faster to Typhoon Morakot than rates of $b$? Even wild speculation of some physical mechanism or process could be enlightening and provide opportunities for future authors to test against field data or models.

489-491: The idea that the typhoon's influence should disappear entirely within a few decades works well with characteristic decay times of 4-9 years, but perhaps not with decay times up to 255 years.

---

## Author Comment (AC1)

*We thank the editor and the reviewers for their insightful and positive reviews. We modified the manuscript following their suggestions and describe all modifications here. In this document,* reviewer comments are in normal font*, and our responses are in* *blue*. *Line numbers in our responses refer to those in the revised manuscript.*

Reviewer 1

Ruetenik and co-authors present a very interesting assessment of multi-year suspended sediment flux data from around Taiwan and assess how suspended sediment discharges and its relationship to hydrology is affected by typhoon Morakot. They compare these data with an inventory of landslides that were triggered by the typhoon and make inferences about the timescales of sediment evacuation from landslides after extreme events. I found this paper to be a very well written and presented contribution that just leaves me with a few suggestions and comments that I invite the authors to consider.

In particular, I wonder about the conclusion that landslides are driving the change in rating curve parameters. You show that many of the catchments have excess sediment yield that is above the landslide yield (by orders of magnitude) – e.g. Figure 8. Does that mean that you are measuring activation of non-landslide parts of the landscape during typhoon Morakot? If you are measuring a substantial proportion of sediment discharge from non-landslide parts of the landscapes, would it be possible that the change in the rating-curve parameters is driven by non-landslide parts of the landscapes just as much as the landslide parts?

Thank you for this suggestion. To point out that non-landslide-derived material may have contributed to suspended sediment fluxes, we added the following text at Line 367. "Where would the excess sediment come from if not from landslides? We hypothesize that a large amount of additional sediment beyond that moved by Morakot-induced landslides was mobilized in the aftermath of Morakot. This is also evident in the slope of the regression in Figure 8, which does not follow a direct 1:1 relationship with landslide intensity. In other words, although basins with greater $I_L$ tend to have greater $\Delta Q_s/A$ values, basins with lower $I_L$ experience proportionally greater erosion rate relative to $I_L$ than basins with larger $I_L$. In particular, roughly half of the basins with $I_L > 10^4$ m$^3$ km$^{-2}$ are below the 1:1 line, suggesting that a large proportion of the excess sediment could be landslide-driven in these basins. Meanwhile, basins on the other side of the 1:1 line are consistent with a portion of non-landslide-derived sources of suspended sediment."

The correlation with landslide intensity could then be due to a co-variation of Typhoon Morakot intensity with landslide intensity. If you plotted Figure 8 and Figure 10 with, for example, rainfall intensity from Morakot instead of landslide intensity, would you find a similar result?

Good question. As you note, it is difficult to tease out these relationships as landslide intensity is inherently correlated with precipitation. To show how landslide intensity varied with Morakot precipitation, we added a plot (S3a) which shows a strong correlation between landslide intensity and Morakot precipitation. In Figure 8, we coloured the data points by Morakot precipitation. We also added supplementary figures (S3b, S4) that plot the same quantities that are on the y-axes in Figures 8 and 10 against Morakot precipitation, as suggested, which show similar correlations to those shown in Figures 8 and 10.

I suggest to give a bit more space to the impact and the implications of looking specifically at suspended sediment transport. In particular, I wonder about the conclusion that the "periods of elevated sediment transport efficiency after landslides should persist from years to decades" (L26-27). Isn't it possible that bedload transport in larger floods will be elevated for many more years? In regards of the discussion of previously measured sediment evacuation times in L36 – 37: As far as I understand, at least some of the references that are cited look specifically at bedload transport (Croissant et al., 2017; Yanites et al., 2010), so the times of export may be quite different.

Thank you for the suggestion. We edited Line 35 to clarify this as follows. "Recent studies have inferred a wide range of recovery times for sediment fluxes responses to landsliding, from several years for suspended sediment fluxes (e.g., Hicks et al., 2008; West et al., 2014; Croissant et al., 2017) to hundreds of years for bedload fluxes (e.g., Yanites et al., 2010)."

Finally, on a minor point, when comparing suspended sediment yield and landslide yield, in Figure 8 and in the associated discussion, I presume the landslide yield includes all grain sizes, so should be a bit lower when compared with suspended sediment yields, right?

We edited Line 320 to emphasize that these represent lower bounds on the total amount of eroded sediment associated with Morakot. "We used the estimates of $Q_s$ at all 87 stations to calculate basin-averaged erosion rate $E$ each year after Morakot. Because these are based on suspended sediment discharge, they do not account for additional mass fluxes as bedload, and therefore should be considered minimum bounds on $E$ (e.g., Dadson et al., 2003)."

I wonder about whether north-south changes in lithology could underly some of the observed north-south trends. My intuition is that this effect should be minor and it is also hard to test for, but maybe worth adding a line about lithology somewhere.

This is an interesting idea. Our intuition is the same as yours – we'd guess that this effect is likely minor. Regardless, in the absence of observations that we could use to test this idea directly, we feel such a discussion would be overly speculative, so we have chosen not to add a discussion of potential lithologic effects.

**Line comments**

L8-9: I can imagine that some readers do not have an intuition of what changes in the coefficient and exponent of the rating curve mean in terms of process (or maybe they do not know what these parameters represent). If there is a way to describe the changes in words and/or define the parameters, that might help. (e.g. instead of saying that the coefficient was a factor of 5 higher, you could say that the suspended sediment transport for a given discharge was a factor of 5 higher etc.)

Thank you for the suggestion. To clarify this, we changed the description in the abstract to the following at Line 10. "Across the compilation of gauging stations, post-Morakot changes in discharge-normalized suspended sediment concentration ($\tilde{a}$) were positively correlated with landslide intensity for seven years after Morakot, while post-Morakot changes in the nonlinearity of the discharge-concentration relationship ($b$) were negatively correlated with landslide intensity from 2011 to 2014."

L17: "Shortly after [...]". Sounds like a repetition of information from previous sentences – may be streamlined.

We changed this to "Furthermore, changes in $a$ and $b$ tended to be larger in basins with more intense landsliding."

L37: With a brief look at the cited reference, I can only see estimates for evacuation times of 250 – 600 years, not thousands of years. Also, as mentioned above, these are, as far as I understand, estimated for bedload transport, not for suspended sediment transport as suggested in this sentence.

Good point. We revised this line in the text to hundreds of years at Line 37.

L110: If eight basins show no landsliding due to Morakot – why not use these eight and instead add the other basins to the group that have landslides?

Our goal in selecting these focus stations is to present two representative groups of geographically distinct stations, one that experienced intense landsliding and another that experienced little to no landsliding. At the four northern focus stations with non-zero landsliding, $I_L$ varied from 4 to 34 m³ km⁻². This is 0.005-0.04% of the mean landslide intensity at the southern focus stations, which is why we grouped these four stations together with the other eight northern focus stations with zero landsliding. To clarify this, we edited the paragraph at Line 104 as follows.

"In Sections 3-4 we present figures from a subset of 24 stations (Figure 1a) that collectively span the range of responses across the full set of 87 stations, and which therefore illustrate the sensitivity of fluvial suspended

sediment fluxes to Morakot-induced landsliding. Twelve of the 24 focus stations (S1-S12 in Figure 1a) are in the southern half of the island, where precipitation during Morakot was highest and where landsliding was most intense. These stations were selected based on the completeness of their monitoring records and because they are in basins with high landslide intensity, and therefore these basins are thought to represent some of the higher responses to landsliding found in the southern part of the island. In these basins, the volume of landslide-mobilized material per unit drainage area ranged from 440 to $2.7 \cdot 10^5$ m³ km⁻². The other twelve stations (N1-N12) are farther north in Taiwan, where precipitation during Morakot was less intense and landsliding was less common. These basins were selected based on the completeness of their monitoring records, and are representative of a typical response for northern stations. Eight of these twelve basins had no Morakot-induced landsliding, and in the remaining four basins, the volume of landslide-mobilized material per unit drainage area ranged from 4 to 34 m³ km⁻². These provide a baseline against which the responses at stations S1-S12 can be compared."

L248 / Figure 4: As far as I understand from the definition in L182, the pre-Morakot values are averaged across the entire period pre Morakot. Here, you have another pre-Morakot value that is just the part of the year 2009 before Morakot (empty circle in Fig. 4) –A different designation for these different measurements would be clearer.

Thank you for pointing out the need for clarity here. To clarify this, we added the following at Line 266. "These values of $a$ and $b$ in the early portion of 2009 are distinct from $a_{pre}$ and $b_{pre}$, which are based on the measurements from all years before 2009."

L296: That sentence doesn't work – maybe the "and" needs to go?

Good catch! We removed this typo.

L337/Figure 8: By plotting the figure in log-space. You are discarding negative values. I wonder what'd happen to the correlation, if you plot it in linear space? Do the catchments with negative values fall on a similar trend?

Plotting the catchments with negative $\Delta Q_s$ values on linear axes does not reveal a strong trend, largely because most of those catchments (63%; 24 of 38) have a landslide intensity of zero, so they would fall along the y-axis of such a plot. We therefore have chosen to leave Figure 8 as is.

L340/Figure9: Can you plot the pre Morakot a-values in the figure, for example as a horizontal/shaded underlay? That would be great to see if the values recover to the previous value or even overshoot.

We added the values of $\tilde{a}_{pre}$ to this figure, as requested, and added the following description at Line 398. "Figure 9 shows that $\tilde{a}$ declines to values approaching $\tilde{a}_{pre}$ within the 11-year time frame at four of the southern focus stations, and it declines to values below $\tilde{a}_{pre}$ at six of the southern focus stations."

L377 – 78: Given the high uncertainties in Fig 10b, I wonder what the likelihood is that there is no difference in the values at low and high landslide intensity. Would it be possible to add a statistical test here?

Good question. The mean recovery times at the low and high landslide intensities are indeed statistically different from one another. We edited Line 378 to show this by reporting the uncertainties on the mean recovery times in the low-$I_L$ and high-$I_L$ groups. "The mean and standard error of $\tau_a$ is 5.8 ± 0.3 years in the basins with the largest landslide intensities ($I_L > 10^5$ m³/km²) and 36.4 ± 14.5 years at landslide intensities smaller than that."

L384 – 386: My understanding of Chen et al. (2020) is that by 2016, the 10Be concentrations are basically not statistically distinguishable from before Morakot when considered across all catchments. Of course, there are some catchments that have lower 10Be than before Morakot but there are also some that have higher concentrations. This sentence suggests to me that the effect of Morakot is still clear in 2016.

Good point. We edited Line 420 to clarify that the effects of Morakot were only apparent in some rivers. "This is consistent with Chen et al. (2020), who observed that ¹⁰Be concentrations in stream sediment were lower in 2016 than they were before Morakot in some rivers and higher in other rivers. This can be interpreted as an indication

that, in some rivers, the Morakot-derived pulse of sediment had not yet been transported away by 2016, while in other rivers it had."

L395: Here and between figures and the text, you switch between $log(\tilde{a})$ and $ln(\tilde{a})$. I presume that $log$ is the log10 and $ln$ is loge? Is there a reason for considering the natural logarithm and sometimes the base 10 logarithm?

We use $\ln(\tilde{a})$ in Figure 9 because the characteristic response time is defined as the reciprocal of the regression slope through $\ln(\tilde{a})$ vs. time, which makes it more convenient to show in this figure. To clarify that we use base-10 logarithms elsewhere in the text, we changed the notation in the text from $log(\tilde{a})$ to $log_{10}(\tilde{a})$ where relevant.

L407/Figure 11: Given the poor fit in panel c, I wonder if the goodness of fit should somehow come into panels b and d?

The standard error of the regression slope (reported directly in Figures 11a and 11c, and shown in the shaded regions in Figures 11b and 11d) is itself a measure of the goodness of fit and therefore already serves this purpose, so we have left this as is.

Also, the "strength of the relationship" sounds a bit like a goodness of fit criterion. Maybe you can find a different wording? For example, "sensitivity of $\Delta log(\tilde{a})$ to landslide intensity.

To clarify, we rephrased this to "The slope between $\Delta log_{10}(\bar{a})$ and $I_L$ gradually decreased over time."

I hope that the comments are helpful, and I remain with best wishes to the authors and editor.

They certainly are! We thank the reviewer for the helpful comments.

Reviewer 2:

**Lines:**
8-10: This is a bit tricky since it's an abstract and necessarily without elaboration, but I'd suggest offering a brief introduction to the coefficients here. This can even take the form of a brief phrase in a sentence, along the lines of " […] the discharge-normalized rating curve coefficient $\tilde{a}$ was higher […] the first year after Morakot (2010), indicating heightened sediment concentrations for given discharges." And similarly for $b$, saying it indicates a greater dependence of sediment concentration on discharge, or an increased sensitivity.

Thank you for the suggestion. We changed this sentence to: "Across the compilation of gauging stations, post-Morakot changes in discharge-normalized sediment concentration ($\tilde{a}$) were positively correlated with landslide intensity for seven years after Morakot, while post-Morakot changes in the exponent of the discharge-concentration relationship ($b$) were negatively correlated with landslide intensity from 2011 to 2014."

10: This could easily just be me, but seeing "Morakot (2010)" made me think it was a citation.

To clarify this, we changed this to "in 2010, the first year after Morakot".

15-16: I think that "Values of $\tilde{a}$ tended to decline faster in basins with more intense landsliding" from 15 is repetitive with "Shortly after Morakot, changes in $\tilde{a}$ and $b$ tended to be larger in basins with more intense landsliding"?

As suggested, we condensed these into one sentence at Line 17. "Values of $\tilde{a}$ increased more and declined faster in basins with more intense landsliding, with a mean characteristic decay time of 6 years in the basins hardest hit by landsliding."

22-27: An effective summary and an interesting result. I appreciate the clarification that just because the increase in sediment transport disappears on a short timescale, that does not mean that the sediment deposited by landslides disappears on short timescales. Not sure that it needs to be in the manuscript, but it does make me wonder where sediment transport rates and sediment availability become detached in these sort of systems. Perhaps it winnows away all of the fine material and becomes transport-limited over short timescales, or maybe much of the landslide material is at elevations inaccessible to the river?

We agree that this is an interesting question. Our data analysis can't answer this directly, but we suspect your suspicion is correct (i.e., that fine sediment is accessible immediately after landslide-derived material is deposited, but is winnowed away over a few years, gradually armoring the bed).

37: To the best of my understanding, Yanites et al. (2010) dealt with bedload transport, not suspended sediment. Additionally, unless you're referring to a different finding, according to Figure 2D the longest timescale for removal of (bedload) material is ~600 years. I'd suggest a different citation for this idea.

Thank you for the suggestion. We edited Line 35 to clarify this as follows. "Recent studies have inferred a wide range of recovery times for sediment fluxes responses to landsliding, from several years for suspended sediment fluxes (e.g., Hicks et al., 2008; West et al., 2014; Crossaint et al., 2017) to hundreds of years for bedload fluxes (e.g., Yanites et al., 2010)."

44-47: Interesting that such a big event only tripled the average annual pre-landslide sediment load. My intuition was that the increase would have been more significant.

To clarify that the cited suspended sediment discharge occurred in *one day*, we edited Line 44 as follows. "Similarly, Typhoon Toraji in 2001 induced >30,000 landslides, which resulted in >175 Mt of suspended sediment discharge in one day from the Choshui River (Dadson et al., 2005), equivalent to three times its annual average sediment load over the period from 1986 to 1999."

85: Based on the manuscript's abstract and introduction so far, I get the impression that the manuscript is specifically focused on analyzing fluvial suspended sediment fluxes, so it might be worth adding that to this line.

Good point! We modified Line 85 to now read: "This manuscript is structured around an analysis of the effects of Morakot-induced landsliding on suspended sediment fluxes."

106-110: This method of splitting up the data between north and south is framed as distinguishing between areas with lower and higher precipitation/landslide intensity, respectively. That's probably a good representation and valid, especially looking at Figure 1. It could be worth quantifying and reporting this here anyways to make this point quantitatively by presenting, for example, average precipitations per group). There may also be other factors beyond precipitation that varied systematically from south to north; Yanites et al. (2018: DOI: 10.1002/esp.4353) focused only on the southern tip, but found or discussed systematic latitude trends in maximum elevation, slope, local relief, channel steepness, erosion/exhumation rates, and channel width.

Thanks for the suggestion! We added violin plots for discharge to the supplement (Figure S1), and we added the following paragraph at Line 117 summarizing the similarities and differences between the north and south focus stations. "The focus stations in the north and south share similar characteristics, with the exception of the rainfall and landslide intensity received during Morakot. For example, the southern and northern focus stations have similar drainage areas (North median = 445 $km^2$, range = 104-1,512 $km^2$; South median = 512 $km^2$, range = 77-2,942 $km^2$) and discharge distributions (Figure S1). Meanwhile, their upstream areas experienced different amounts of precipitation during Morakot (North median = 0.47 m, range = 0.25-0.89 m; South median = 1.31 m, range = 0.63-2.03 m), as well as landslide intensity (North median = 0 $m^3/m^2$, range = 0 - 3.4 × $10^{-5}$ $m^3/m^2$; South median = 0.04 $m^3/m^2$, range = 4.42 × $10^{-4}$ - 0.27 $m^3/m^2$)."

Figure 1: I find it a bit challenging to identify all of the small red dots because of their size, having the same color as the large red dots, and sometimes being apparently overlain by other dots. Perhaps enlarging them and

using a third color could help. Additionally, I could see some readers wishing the colored contours using a perceptually uniform colorbar so as to not artificially introduce breaks, as jet tends to do [https://colorcet.com/].

To improve this, we changed the colors of the dots in Figure 1a and made them larger, and we also changed the color map for precipitation to one that is perceptually uniform.

126-127: Here is a nice, concise introduction to the meaning of the parameters; some version of this also mentioned above would be helpful.

Thank you. We have now added this explanation to the abstract at Line 10. "Across the compilation of gauging stations, post-Morakot changes in discharge-normalized sediment concentration ($\tilde{a}$) were positively correlated with landslide intensity for seven years after Morakot, while post-Morakot changes in the exponent of the discharge-concentration relationship ($b$) were negatively correlated with landslide intensity from 2011 to 2014."

131: I came back to this line after reaching section 2.3 and feeling that landsliding intensity had not been emphasized earlier in the methods. While it's mentioned in the Introduction and shown in Figure 1, I think it's worth making sure that you clarify that you are measuring the influence of Morakot-induced landsliding, and not Morakot itself (ex. precipitation). This fits better with the title of the paper and introduction. I'd also look at mentions in section 2.2 and the caption of Figure 3 to ensure you properly emphasize that you are estimating the effects of landsliding (due to Typhoon Morakot) on suspended sediment discharges.

We clarified on this at line 141: "Our goal is to quantify the influence of Morakot-induced landsliding on suspended sediment transport by tracking the evolution of $Q_s$ and the rating curve parameters over time." We also clarified this at line 202: "We aim to quantify the effects of Morakot-induced landsliding on suspended sediment loads over the decade after Morakot." We have left Figure 3's caption as is, since the point of Figure 3 is to demonstrate differences in pre-Morakot and post-Morakot rating curves, not to interpret these differences as a result of landsliding.

134: Consider removing "applying".

Done.

135: I think that readers who have not constructed rating curves may not understand what centering means in an intuitive sense. Is it akin to horizontally or vertically shifting values in some log-space? It looks like it might be normalization, or comparison of values as deviations from a central value. A few words would be appreciated, especially since it seems Cohn et al. (1992) do not provide much in the way of explanation themselves.
143: Ahh! It seems this line contains the graphical explanation I was looking for above.

To clarify this, we added a reference to Figure 2 in this description on line 145.

151: To each year's C and Q data for each station, right?

Correct, clarified.

161: "sediment load" -> "suspended sediment load"

Done.

166: Where/how did you get or measure A for each gauging station? It isn't mentioned as part of the WRA dataset on lines 95-96. I'm assuming a DEM and some routing toolkit, but it bears mentioning.

Good point – this should be included in the Methods. We used MERIT Hydro and its associated flow direction grid. To clarify this, we added the appropriate citation (Yamazaki et al., 2019) to the Methods.

169-170: A good disclaimer. I think it would be worth a half-sentence about how far off you might expect the suspended sediment load to be from the total sediment load. Is suspended load 10% of the total load? 50%, 90%?

We added the following at Line 182 to address this. "For context, Dadson et al. (2003) estimated that bedload from high mountainous rivers could account for $30 \pm 28$ % (95% CI) of the total sediment discharge."

Eqn (5): Should E's subscript be 1 instead of 3, since it is associated with downstream gauging station 1?

This was indeed a typo. We corrected it.

179-180: You say here that you apply Eqn's 1-5 to each gauging station's measurements each year, to yield estimates of parameters at each gauging station. However, in sections 3.1 and the first two paragraphs of 3.2, it seems that you only present the results (and statistics) for estimates of $a$, $b$, and $Qs$ for the 24 focus stations. Then, for the second half of 3.2, you say you calculated $Qs$ for all 87 stations and use them to calculate and present $E$ for all all 87 stations. I'm unclear as to the selection criteria for the focus stations and why only 24 were presented and used for summary statistics, if it appears that values were calculated for all 87. Do we believe that the focus stations are representative of all stations?

Correct: We calculated these parameters for all 87 stations and report them in the supplementary tables. In the interest of brevity in the main text, we show results from a representative subset of 24 stations, as stated when the focus stations are introduced in Section 2. To clarify this, we expanded on this at Line 104 as follows. "In Sections 3-4 we present figures from a subset of 24 stations (Figure 1a) that collectively span the range of responses across the full set of 87 stations, and which therefore illustrate the sensitivity of fluvial suspended sediment fluxes to Morakot-induced landsliding. Twelve of the 24 focus stations (S1-S12 in Figure 1a) are in the southern half of the island, where precipitation during Morakot was highest and where landsliding was most intense. These stations were selected based on the completeness of their monitoring records and because they are in basins with high landslide intensity, and therefore these basins are thought to represent some of the higher responses to landsliding found in the southern part of the island. In these basins, the volume of landslide-mobilized material per unit drainage area ranged from 440 to $2.7 \cdot 10^5$ $m^3$ $km^{-2}$. The other twelve stations (N1-N12) are farther north in Taiwan, where precipitation during Morakot was less intense and landsliding was less common. These basins were selected based on the completeness of their monitoring records, and are representative of a typical response for northern stations. Eight of these twelve basins had no Morakot-induced landsliding, and in the remaining four basins, the volume of landslide-mobilized material per unit drainage area ranged from 4 to 34 $m^3$ $km^{-2}$. These provide a baseline against which the responses at stations S1-S12 can be compared."

188: The tense elsewhere has been past, but is present here. This may be intentional but it's worth giving a look-over to ensure consistency.

Good catch! We changed the sentences referred to here to past tense.

188-193: The first sentence is a good topic, but the rest of this paragraph seems a bit out-of-place. It might work if you add what's missing after the first sentence: a connection that says we are aiming to quantify the effects of Morakot by using changes in rating curves before and after a given storm. Then the following lines justify that it's reasonable to anticipate seeing these changes in rating curves; in fact, we should expect to.

We agree this is a worthwhile addition. We edited Line 202 to read: "We aim to quantify the effects of Morakot-induced landsliding on suspended sediment loads over the decade after Morakot. For rivers heavily affected by landsliding, this necessitates quantifying changes in rating curves before and after Morakot."

194-215: This is a nice and clear explanation of why average Qs rates pre- and post-storm aren't sufficient to answer the research question. Only note is that you may be able to combine the final two sentences (213-215) into the paragraph above (208-212).

We integrated these sentences into the previous paragraph as suggested.

222-223: Marc et al. (2018) is cited twice in the same phrase.

Thanks for catching that! Fixed.

222-224: "in which it is assumed" as used here applies to "corrections", and not the catalogue estimates themselves. Is this intentional?

To clarify this, we modified this sentence at Line 236 as follows. "In the landslide catalogue in Marc et al. (2018), the calculation of $V_L$ accounts for amalgamated landslide polygons, where it is assumed that each landslide has an elliptical shape and a mean width calculated with the formula proposed and validated by Marc et al. (2018)."

I'd also define c and p here; you can still mention the values later in the paragraph, though if you did not do the estimations yourself (as you say on line 228) you might not need to mention the explicit values of p and c here.

We edited Line 236 to clarify the parameters as suggested: "where the sole parameters are the scaling coefficient c and exponent p."

229: Insert "for each station"

Done.

232: I think section 3 should include a summary of the differences between the North and South sections in terms of the "input" conditions, to communicate to the reader that they are comparable in all manners (drainage areas, typical discharges) except for the treatment condition (precipitation and landsliding intensity). Much of the following results relies on them being distinct in terms of precipitation and landsliding intensity, and presumably not in other ways. As such, why not include a figure with a couple of box-and-whisker or violin plots showing that there is a quantitative difference between the two groups in terms of precipitation and landslide intensity and how significant the difference is or isn't? That would make statements like the first sentence of Figure 4 caption easier to justify and write, since it will be in readers' heads already.

This suggestion is similar to the reviewer's suggestion at lines 106-110, so here we use the same response we posted to that comment above.

We added violin plots for discharge to the supplement (Figure S1), and we added the following paragraph at Line 117 summarizing the similarities and differences between the north and south focus stations. "The focus stations in the north and south share similar characteristics, with the exception of the rainfall and landslide intensity received during Morakot. For example, the southern and northern focus stations have similar drainage areas (North median = 445 km², range = 104-1,512 km²; South median = 512 km², range = 77-2,942 km²) and discharge distributions (Figure S1). Meanwhile, their upstream areas experienced different amounts of precipitation during Morakot (North median = 0.47 m, range = 0.25-0.89 m; South median = 1.31 m, range = 0.63-2.03 m), as well as landslide intensity (North median = 0 m³/m², range = 0 - 3.4 × 10⁻⁵ m³/m²; South median = 0.04 m³/m², range = 4.42 × 10⁻⁴ - 0.27 m³/m²)."

234-235: refer to Figure 4

Done.

242-244: Is there a figure showing this anywhere? Could be simple and useful to communicate the differences between the two zones. Either as two box-and-whisker plots, or with station # on the x-axis and $\tilde{a}/a_{pre}$ on the y.

We prefer to cite these values in the supplementary table rather than add a new figure, so we added a citation to the supplementary table on this line.

252: This is entirely up to personal style, but if one wanted to avoid rhetorical or prompting questions and reserve them for research questions or prompting discussion sections, one way to re-write this could be "While this could account for the observation […] at some stations, we consider this unlikely."

As suggested, we changed the wording to "While this could account for the observation that $a$ is higher during post-Morakot 2009 than it was before Morakot at some stations, we consider this unlikely."

252-257: This is a good argument, but I'm not sure it's necessary unless you expect readers to bring it up unprompted. Could be shortened if you wanted, though I always respect including disclaimers. On the other hand, if it were very important to prove this, you could do a similar calendar split on each of the pre-Morakot years and see how common or uncommon such a within-year jump in $a$ is.

Thank you for the suggestion. We prefer not shortening this paragraph, so we have left this text as is.

Figure 4 caption: The final sentence is a result, and is not shown in this figure, so I do not believe that this idea should first appear here unless it is accompanied by a reference to an associated supplemental figure.

We added the stars to show that the largest flood events do not always correspond with the largest changes in $a$ and $b$. To clarify this, we removed the final sentence from the caption, as suggested, and added the following to line 510. "Furthermore, at many stations, the largest recorded flood event during 1990-present occurred in years other than 2009 (white stars, Figure 4). This suggests that Morakot-induced changes in $\tilde{a}$ and $b$ were likely driven by changes in landslide-derived sediment supply, not changes in $Q$."

Is it shown or discussed elsewhere? Additionally, I'm not sure that the word "confirming" in the phrase "[…] confirming that basins with minimal landsliding experienced smaller changes in rating curves" is the most appropriate, as I'm not sure we've seen quantitative evidence yet that the northern and southern stations differ systematically in landsliding intensity beyond that shown visually in Figure 2. It almost looks like it might correlate similarly well to precipitation intensity.

We agree that "confirming" was too strong a word and that readers should be directed to the data showing the differences in landsliding intensity between the northern and southern focus stations, so we edited the caption as follows. "By contrast, at the northern focus stations, where landslide intensity was smaller (Table S2), $\tilde{a}$ and $b$ show smaller responses to Morakot."

Section 3.1: This might be personal taste as well, but I'd consider including a table of summary results/statistics between the north and south areas. The table could present many of the average pre-Morakot and post-Morkot 2009/2010 values for the different parameters of interest ($\tilde{a}$, $b$, $Qs$). That would replace much of the writing, but I suppose that might make this section quite brief, and might not work for comparisons that rely on looking at values in 2011 (such as $b$ in the south), so might not be the effort.

We appreciate the suggestion. The supplementary table already contains this information, and we believe there would be too much information to fit into a regular table in the main text, so in the interest of brevity and avoiding redundancy, we have chosen to leave the text as is to refer readers to the supplement.

258-270: I think it might be helpful to put the magnitude of these changes into context for readers who are not as familiar with Taiwan or the parameters. It was not as big of an issue with values of $\tilde{a}$, where changes were on a factor of ~2-8x. Here, all we can see is that the changes at the southern stations from pre-Morakot to post-Morakot 2009-2010 were much smaller than the changes from pre-Morakot to 2011, and the northern changes from pre-Morakot to post-Morakot 2010 are somewhere in the middle. What we lack is an understanding of the relative scale between the largest changes we see here compared and the sort of changes over comparable timescales that would be expected from normal annual variations without a typhoon. An easy way to do this is to compare the magnitude of these changes to, say, the standard deviation of the values pre-Morakot or the average magnitude of changes between years. Essentially, when we see an average year-over-year drop from 0.59 to 0.29, is that an exceptionally strong signal, or is that within the range of noise we could see in pre-Morakot measurements and thus indistinguishable from random chance? It should be possible to do some sort of basic

statistical test to see which of the changesassigned to Morakot are significant vs insignificant, given the pre Morakot history in annual rating curve parameters.

Thank you for the suggestion. We added some key statistics to put the changes in *b* in context at Line 285: "To put these changes into perspective, the average *b* values in 2009 and 2010 were in the 54th and 58th percentiles compared with the historical averages in the North and South focus stations, respectively. From 2011 to 2015, the average *b* value in the southern focus stations was in the 21st percentile of their respective historical values, while the Northern stations were on average in the 54th percentile. Thus, persistently lower values of *b* appeared in several basins with intense landsliding, but not until the second year after Morakot."

281-283: Above, you compared post-Morakot values of rating curve parameters to the mean pre-Morakot values, but here you compare post-Morakot values of *Qs* to the median pre-Morakot values. Any particular reason for the different method?

This appears to be a misunderstanding.  At each station, the value of $a_{pre}$ is based on a single rating curve applied to all pre-Morakot data. It is not the mean of all annual values of *a* in each year before Morakot. We modified Line 194 to clarify this. "We also applied this method to the entire period of *C* and *Q* measurements before Morakot at each gauging station to calculate the rating curve parameters based on all pre-Morakot data, which we denote $a_{pre}$ and $b_{pre}$ (Supplementary Table S2)."

283-284: I think I know what you mean, but saying "rapid drop-off in *Qs* after Morakot" almost sounds like it declined after Morakot. I think it'd be more accurate to say that the drop-offs occurred after the post-Morakot peaks. Additionally, here (as well as the caption for Figure 4) a reference is made to changes in annual discharge values. Are the timeseries of discharge, or even just annual averages per-station or for north vs south, shown anywhere?

Thank you for the suggestion. We plotted the maximum event magnitude by year in a supplementary figure (Figure S2), which we now cite here. This shows that the highest peak in *Q* on record at many of the southern focus stations occurred in 2009.  By contrast. the peak *Q* in 2009 is not the highest on record for most of the northern focus stations.

287-279: I'm not sure if the "all but one" station mentioned in the first line is station S12 mentioned in the second line, or if these two lines are related to one another.

Good point. To clarify this, we rephrased this as follows to emphasize that these are two related statements. "With the exception of this station, which had a higher sediment load in 2010 than 2009 post-Morakot, the annual sediment discharge at all stations in both 2010 and 2011 was less than 25% of the sediment discharge in August-October 2009."

Figure 5: Interestingly, N1-3 have increases in suspended sediment discharge that look like those of the southern stations. I wondered if N1-3 are the south-most of the Northern stations. Based on Figure 2, it looks like these stations are southern, but not as much as N11/12. It also looks like these stations did experience elevated precipitation compared to the other Northern stations, but N2 and N3 experienced no landslides. This leads one to wonder if some of the increased *Qs* observed is due to non-landslide associated mechanisms that also correlate to precipitation. It would be interesting to regress *Qs* separately against landslide intensity and against precipitation.

As suggested, we added a plot of excess $Q_s$ against basin-averaged Morakot precipitation and added it to the supplement as Figure S3b.  Additionally, to show how landslide intensity varies with Morakot precipitation, we added Figure S3a which shows a strong correlation between landslide intensity and Morakot precipitation. This shows that it is difficult to tease out these relationships.  Additionally, in Figure 8, we coloured the data points by Morakot precipitation, which shows that the highest landslide intensities tended to occur in basins with the highest Morakot precipitation.

Figure 6: How are these basins delineated? Is there a DEM involved? Is it from the WRA dataset?

We edited the Methods section to state that these are based on MERIT (unidirectional) flow routing, such that 176 now reads: "After computing $Q_s$ at each gauging station, we computed basin-averaged erosion rates $E$ [L T$^{-1}$] by dividing $Q_s$ by the drainage area $A$ upstream of the gauging station and an assumed bedrock density $\rho_r$ of 2700 km m$^{-3}$, where drainage areas and flow-routing information were extracted from MERIT Hydro (Yamazaki et al. 2019) and used in subsequent calculations."

295-307: I think that these results should be in a section of their own, and not part of "3.2 Suspended sediment discharge $Qs$". Additionally, though it was mentioned in the methods, it bears repeating here that this is basin-averaged erosion or deposition rates associated with suspended sediment, specifically (as opposed to total actual erosion which presumably would be higher as it included bedload, etc.).

Excellent point! As suggested, we put these results in a new section "3.3: Annual erosion rate estimates". We also re-emphasized that these are minimum erosion rates at Line 323: " Because these are based on suspended sediment discharge, they do not account for additional mass fluxes as bedload, and therefore should be considered minimum bounds on $E$ (Dadson et al., 2003)."

296: remove "and"

Done.

297: Is this station better-known as S# or N#? I suppose not, if it's not one of the focus stations.

Correct: it is not one of the focus stations. We therefore referred to it by its code in the WRA database.

298: How does this compare to "normal", non-Typhoon annual basin-average erosion rates for this type of setting?

We added the following to address this at line 326: "For example, the sediment discharge from the small basin above station 1660 H010 on the Erhjen River in the post-Morakot portion of 2009 is equivalent to >10$^3$ mm of basin-averaged erosion, well above the pre-Morakot median of 25 mm/yr."

Figure 7: Missing the word "of" in "By contrast, at seven of the twelve northern stations […]"

Fixed.

325-337: I like this presentation, and I also think it's a reasonable result to find that basins that over-deliver sediment have lower landslide intensities than those that under-deliver. I suppose this suggests that the range of $Qs$ amounts delivered by rivers exists over a narrower range than that of landslide intensities, and that the former is only moderately sensitive to the latter. Have you tried this same regression against rainfall intensity (which also has units of L3/L2)? That might give some insight as to where some of the non-suspended sediment is coming from. It's also interesting that in most cases, $\Delta Qs$ is greater than the volume of landslided material considering the former doesn't include bedload transport, and I imagine a lot of the landslide material is coarser than would be expected from suspended sediment. This suggests to me that either landslides somehow "prime" landscapes to deliver excess sediment, or much of this excess suspended sediment you observe is not, in fact, material that was captured in the 2018 landslide dataset. Do you have a few words about what the character of this material might be, or the process that mobilizes it?

Thank you for the suggestion. At Line 367 we added the following to emphasize that some of the suspended sediment may be derived from non-landslide sources.

"Where would the excess sediment come from if not from landslides? We hypothesize that a large amount of additional sediment beyond that moved by Morakot-induced landslides was mobilized in the aftermath of Morakot. This is also evident in the slope of the regression in Figure 8, which does not follow a direct 1:1 relationship with landslide intensity. In other words, although basins with greater $I_L$ tend to have greater $\Delta Q_s/A$

values, basins with lower $I_L$ experience proportionally greater erosion rate relative to $I_L$ than basins with larger $I_L$. In particular, roughly half of the basins with $I_L > 10^4$ m$^3$ km$^{-2}$ are below the 1:1 line, suggesting that a large proportion of the excess sediment could be landslide-driven in these basins. Meanwhile, basins on the other side of the 1:1 line are consistent with a portion of non-landslide-derived sources of suspended sediment."

Figure 8: The caption says what is shown by each point, but not what each point is. Is each one of the 24 focus stations?

The data points in Figure 8 are all stations with positive $\Delta Q_s$ and $I_L$ values, not the focus stations. To clarify this, we added "for all stations with positive $\Delta Q_s$ and $I_L$ values (n = 24)" to the first sentence of the figure caption.

Figure 9: I'd recommend adding a horizontal line showing the pre-Morakot value for reference. Also, are these the same data as are shown in Figure 4? If so, any particular reason/justification for using $\log(\tilde{a})$ there (and later in section 4.2) and $\ln(\tilde{a})$ now?

We use $\ln(\tilde{a})$ in Figure 9 because the characteristic response time is defined as the reciprocal of the regression slope through $\ln(\tilde{a})$ vs. time, which makes it more convenient to show in this figure. To clarify that we use base-10 logarithms elsewhere in the text, we changed the notation in the text from $\log(\tilde{a})$ to $\log_{10}(\tilde{a})$ where relevant. We added horizontal lines to Figure 9, as suggested.

377-380: The question and content asked and described here are presented earlier in the paragraph above, on lines 368-373. I'd suggest rearranging or combining.

Thank you for the suggestion! We combined these paragraphs, as suggested.

Figure 10: I am thinking about how to compare the values shown here in b) to the take-away from lines 365-359, where the minimum decay timescale was ~4 years and the maximum was ~ 9 years. It seems that many here are >9 years.

These lines in the text specify that the response time is 4-9 years "at the ten southern stations that have declining values of a after Morakot". By contrast, Figure 10b includes all the stations with negative regression slopes. To clarify this, we added the following sentence at Line 403. "We also calculated $\tau_a$ for the 26 stations at which $a$ declines after Morakot to obtain characteristic decay times."

383-391: This argument is reasonable, and I think it makes sense. It could probably be validated by looking at modern satellite images. However, I am not sure how to rectify it with the data shown in Figure 8, which shows that many/most of the stations have already delivered more excess sediment (purely via suspended load) than the total landslided volume. The volume of delivery would presumably be even larger if bedload were included. I think this should be acknowledged and/or discussed somewhere.

Agreed. We added text stating that these are based on suspended sediment fluxes and thus are lower bounds on total erosion, as in our response to Line 295 above.

482: I believe that this is the first time that the characteristic decay time of 3-255 years has been presented, at least in words. Should be presented before the conclusions section. I assume that this is the total range among all stations, and the 4-9 years presented before is among a subset of stations (focus, or high-landslide-intensity and also decaying, etc.).

Agreed. To clarify this, we modified Line 522 to emphasize that 3-255 years is the range for all stations, and 4-9 years is the range for the 10 southern stations with negative regression slopes. "This was followed by a decline in $a$ with an exponential characteristic decay time of 3-255 years for all stations, with shorter (sub-decadal) decay times in basins with more intense landsliding (4-9 years for our southern focus stations)."

483-484: Do you have any intuition or understanding of why rates of $\tilde{a}$ should respond faster to Typhoon Morakot than rates of $b$? Even wild speculation of some physical mechanism or process could be enlightening and provide opportunities for future authors to test against field data or models.

On the contrary, Figures 11b and 11d show that changes in $a$ and $b$ lose their sensitivity to landslide intensity over about the same time scale. We prefer not to speculate wildly, so we have otherwise left the text as is.

489-491: The idea that the typhoon's influence should disappear entirely within a few decades works well with characteristic decay times of 4-9 years, but perhaps not with decay times up to 255 years.

As suggested, we edited Line 529 to emphasize that the sub-decadal decay times occur in the basins with the most intense landsliding. "Together, these observations are consistent with an influence of landsliding on suspended sediment transport efficiency that was large immediately after Morakot and then diminished rapidly in most basins. This implies that in the basins that experienced the heaviest landsliding, the influence of Morakot-induced landsliding on suspended sediment concentrations substantially declined within the first decade after the typhoon and that its influence will disappear entirely within a few decades."